# C²INet: Realizing Incremental Trajectory Prediction with Prior-Aware Continual Causal Intervention

## Abstract

Trajectory prediction for multi-agents in complex scenarios is crucial for applications like autonomous driving. However, existing methods often overlook environmental biases, which leads to poor generalization. Additionally, hardware constraints limit the use of large-scale data across environments, and continual learning settings exacerbate the challenge of catastrophic forgetting. To address these issues, we propose the Continual Causal Intervention (C²INet) method for generalizable multi-agent trajectory prediction within a continual learning framework. Using variational inference, we align environment-related prior with the posterior estimator of confounding factors in the latent space, thereby intervening in causal correlations that affect trajectory representation. Furthermore, we store optimal variational priors across various scenarios using a memory queue, ensuring continuous debiasing during incremental task training. The proposed C²INet enhances adaptability to diverse tasks while preserving previous task information to prevent catastrophic forgetting. It also incorporates pruning strategies to mitigate overfitting. Comparative evaluations on three real and synthetic complex datasets against state-of-the-art methods demonstrate that our proposed method consistently achieves reliable prediction performance, effectively mitigating confounding factors unique to different scenarios. This highlights the practical value of our method for real-world applications.

## 1 Introduction

Predicting the movement trajectories of multiple agents across various scenarios is a critical challenge in numerous research areas, including autonomous driving decision systems (Gao et al., 2020; Shi et al., 2024), security surveillance systems (Xu et al., 2022b), traffic flow analysis (Zhang et al., 2022), and sports technology (Xu et al., 2023; 2022a). Trajectory sequence-based representation learning has demonstrated considerable efficacy in addressing this challenge (Gupta et al., 2018; Yu et al., 2020; Shi et al., 2021). Typically, partial trajectories of observable agents serve as input, encoded by integrating spatial positions, inter-agent interactions, and scene semantics. This process generates a latent representation of the observed trajectories, enabling the prediction of future paths through probabilistic inference or recurrent decoding methods. Many existing studies focus on training and prediction within specific scenarios (Yuan et al., 2021). For example, predictions made on a low-traffic highway often favor rapid, straight-line movements. In contrast, predictions in high-traffic urban environments require models to account for irregular routes, as vehicles frequently avoid obstacles. Furthermore, regional regulations and customs can significantly affect trajectory prediction. On roads where motorized and non-motorized vehicles share lanes, regional driving styles influence trajectory patterns, including the spacing between agents and their speed. A model lacking generalizability cannot be effectively applied across diverse scenarios, necessitating significant computational resources for retraining. Moreover, trajectory prediction is critical in applications like autonomous driving and security surveillance, where failures can have catastrophic consequences.

These issues primarily arise because multiple interacting factors influence the trajectory paths of agents. Empirical samples often contain spurious features, which can be incorporated into conditional probability models, leading to neither directly observable nor explainable confounders. These confounders distort representations and prediction outcomes. Moreover, the intrinsic trajectory patterns may shift significantly when inferring in a new domain. Eliminating confounding biases during training does not ensure their absence during inference. Two key challenges contribute to this: the incorrect introduction of past intervention factors into new causal relationships and the failure to account for newly emerging relevant factors. Unlike image data with distinct categories, trajectory data in spatial contexts exhibit infinite variations, complicating the establishment of feature-scene relationships and rendering traditional bias elimination methods ineffective (Zhang et al., 2020).

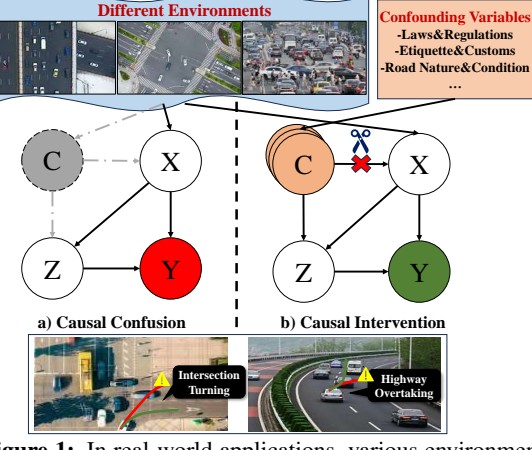

**Figure 1:** In real-world applications, various environmental scenarios often contain confounding variables (denoted as $C$), such as regulations, customs, and road conditions, influencing trajectory data. To address this, our method constructs a causal model designed to mitigate the effects of spurious correlations on prediction outcomes. For instance, predicting a turn at an intersection and a lane change on a highway may be conflated, as both trajectories exhibit an initial directional shift.

Causal inference techniques have been developing rapidly in recent years, primarily used to eliminate factors that interfere with outcomes and to study the direct relationships between phenomena. Based on causal intervention techniques, it is possible to eliminate confounding biases in the complete paths of multiple agents, which affects the trajectory generation process and directly interferes with the predictor through biased pathways. A typical backdoor adjustment method blocks the interference of confounding variables on causal effects. By estimating the intervention distribution, the true effects influencing the trajectories are captured. However, existing methods focus exclusively on source-specific and stationary observational data. These learning strategies assume that all observational data is available during the training phase and comes from a single source (Yuan et al., 2021; Kamenev et al., 2022). In rapidly changing real-world applications, this assumption no longer holds. For example, in autonomous driving, the surrounding vehicles and pedestrians are constantly changing. To ensure accurate behavior prediction, new environmental data must be continuously incorporated during training. This requires designing perception-related factors with a strong capability to handle temporal variations in time-series data. Additionally, in ever-changing scenarios, models often face catastrophic forgetting. Due to privacy protection or limited storage space, models can not have access to complete historical data during training. Therefore, it is essential to ensure that old correlations are remembered during continuous training.

This paper investigates the widely adopted end-to-end trajectory representation paradigm to enable continuous intervention in prediction models. This approach encodes observed data $X$ to estimate the probabilistic distribution of the latent representation $P(Z|X)$, and then predict future trajectories $P(Y|Z)$. We identify a confounding factor $C$ that influences the distribution of observed trajectories $X$, the latent representation $Z$, and the predicted trajectories $Y$. Building on this, we propose a method for continuous intervention in the latent representation. First, we apply the $do$-operator to intervene in the observed trajectories $X$, reducing the impact of spurious environmental factors (e.g., traffic rules, social norms) on the feature distribution. Unlike previous methods, our approach does not require prior exposure to additional data to achieve stable predictions. Second, we introduce a progressive continuous training strategy that ensures the trajectory representations adapt to newly acquired domain. Third, extensive experiments show that our model effectively handles incremental trajectory samples while mitigating catastrophic forgetting, leading to improved bias elimination.

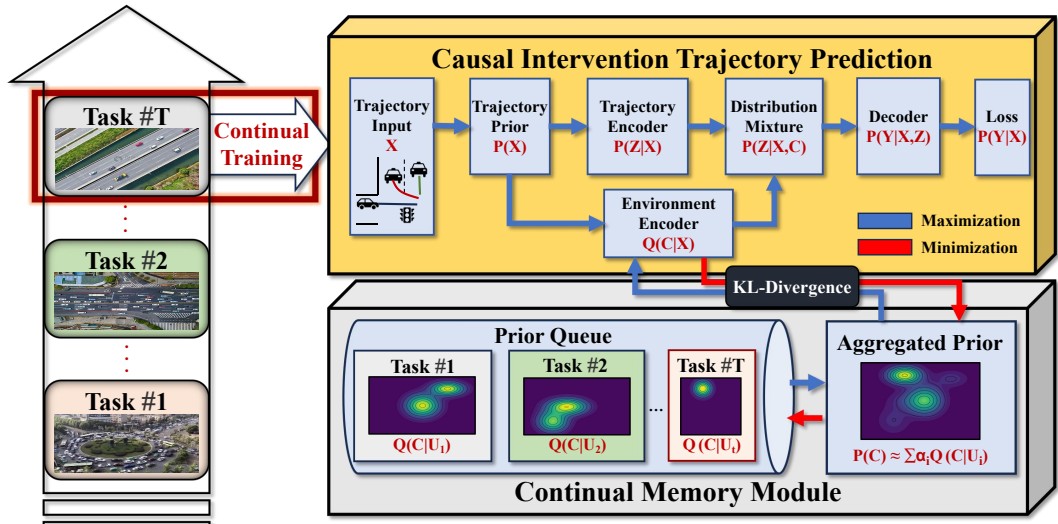

Figure 2: The proposed C$^2$INet model incorporates a causal intervention trajectory prediction framework and a Continual Memory module with a prior queue. Utilizing a min-max training strategy, the model is optimized while acquiring optimal continual prior for newly added scenarios.

## 2 CAUSAL INTERVENTION FOR TRAJECTORY PREDICTION

### 2.1 PROBLEM FORMULATION

The multi-agent trajectory prediction problem primarily focuses on scenarios with $m$ agents moving simultaneously within a certain time $n$ frames. Their trajectories are denoted as $T = \{t_1, t_2, \ldots, t_m\}$. The trajectory of the $i$-th agent is defined as $t_i = \{x_1, x_2, \ldots, x_n\}$, where $x_i \in \mathbb{R}^3$ or $x_i \in \mathbb{R}^2$ represents the coordinate position. A typical trajectory prediction task involves predicting the future trajectory $Y_i = \{x_{\text{obs}+1}, x_{\text{obs}+2}, \ldots, x_n\}$ based on the observed trajectory $X_i = \{x_1, x_2, \ldots, x_{\text{obs}}\}$. Assuming the trajectory data is collected from different task domains, such as pedestrians and vehicles in various scenarios across different cities, it can be represented as $\{D_1, D_2, \ldots, D_N\}$. This work focuses on sequential learning across domains without accessing task/domain information. During training, each task domain is loaded sequentially as a data stream, with the model processing only the current task's samples $D_K$ in small batches. For testing, the model's performance is validated on previously encountered domains $\{D_1, D_2, \ldots, D_K\}$ by computing the empirical probability distribution and generating prediction results. The contextual information is computed as $E_i = E(T_i, C_i)$, where $C_i$ represents the background characteristics of the $i$-th environment and $T_i$ denotes the trajectory patterns, reflecting the interactions between agents and the environment. The trajectory prediction problem is then formulated as $Y = F(X, E)$.

### 2.2 CAUSALITY ANALYSIS

We first define the causal variables related to trajectory representations and prediction models, which are critical for constructing the Causal Structure Model (CSM). In the CSM, $X \rightarrow Y$ signifies that changes in $X$ directly impact $Y$, indicating a direct causal relationship. Following our previous setup, we introduce the confounding factor $C$, a contextual prior representing environmental information and influencing trajectory generation. For instance, acquaintances may walk side by side or pause when interacting, while strangers typically avoid each other. Similarly, turning trajectories differ depending on whether driving occurs on the left or right side of the road, varying by region.

Further, we construct a causal graph to represent the dependency relationships between trajectory data distribution and the prediction model as shown in Fig.1. The causal path $C \rightarrow X$ illustrates the influence of environmental prior on trajectory patterns, a direct and easily defined relationship. Given that the model includes an encoder $P(Z|X)$, which extracts features $Z$ from observed trajectories $X$, we establish the dependency path $X \rightarrow Z \leftarrow C$. The influence of $C$ on $Z$ stems from

the encoder, which integrates environmental, map, and interaction information. While this enhances performance, it also introduces harmful associations that affect accuracy.

The final prediction is generated by the decoder $P(Y|Z)$, which computes future trajectory distributions or derives specific positions from the features of the known trajectories, leading to the dependency $X \to Y \leftarrow Z$. Analyzing the CSM, we find that the confounding effect of $C$ influences both the generation of known trajectories $X$ and the representation $Z$ (and by extension, $Y$). This confounding distorts the model's ability to capture accurate causal relationships, complicating the exploration of intrinsic properties in multi-agent trajectory data.

## 2.3 PREDICTION INTERVENTION

We introduce a causal intervention framework using the $do$-operator $P(Y|do(X))$ to intervene in the conditional probability and mitigate confounded causal relationships. This intervention removes the direct influence of $C$ on $X$, as illustrated in Fig.2. Based on the Markov probability theorem, beneficial interventions render $Z$ independent of the confounding disturbance from $C$. The formula $P(Y|do(X))$ intervenes in $X$, effectively cutting off the path $C \to X$ and establishing the direct relationship $X \to Z \to Y$. However, environmental influence are often inaccessible in real-world scenarios, and collecting comprehensive trajectory data is costly and complex. In this work, we apply the backdoor adjustment method, assigning sampled values to $C$, simulating trajectory probability relationships under different environments while removing confounding effects by cutting off the path $C \to X$. Specifically, we aim to solve the following formula:

$$
\begin{aligned}
P(Y|do(X)) &= \mathbb{E}_{\substack{P(Z)\\P(C)}} P(Y|do(X), Z, C)\\
&= \mathbb{E}_{\substack{P(Z)\\P(C)}} P(Y|do(X), Z, C)P(Z|do(X), C)P(C|do(X)).\\
&= \mathbb{E}_{\substack{P(Z)\\P(C)}} P(Y|X, Z, C)P(Z|X, C)P(C) = \mathbb{E}_{\substack{P(Z)\\P(C)}} P(Y, Z, C|X)\\
&= \mathbb{E}_{\substack{P(Z)\\P(C)}} P(Y, Z|X)P(C) = \mathbb{E}_{P(Z)} \frac{P(Y, Z, C|X)}{P(C|X, Y, Z)}.
\end{aligned}
\tag{1}
$$

The derivation details of formula $P(Y|do(X))$ can be found in App. A.2.1. In this context, directly accessing $P(C)$ is challenging, making it infeasible to compute $P(Y|do(X))$ directly. Using the log-likelihood function for maximum likelihood estimation, Eq.1 are transformed as follows:

$$
\begin{aligned}
\log P(Y|do(X)) &= \mathbb{E}_{\substack{P(Z)\\Q(C|X)}} \log \frac{P(Y, Z, C|X)}{P(C|X, Y, Z)}\\
&= \mathbb{E}_{\substack{P(Z)\\Q(C|X)}} \log \frac{P(Y, Z, C|X)Q(C|X)}{P(C|X, Y, Z)Q(C|X)}\\
&= \mathbb{E}_{\substack{P(Z)\\Q(C|X)}} \log \frac{P(Y|X, Z, C)P(Z|X, C)P(C)Q(C|X)}{P(C|X, Y, Z)Q(C|X)}\\
&= \mathbb{E}_{Q(C|X)} \log \frac{P(Y|X, Z, C)P(C)Q(C|X)}{P(C|X, Y, Z)Q(C|X)}\\
&= \mathbb{E}_{Q(C|X)}[\log P(Y|X, Z, C)] + \mathrm{KL}(Q(C|X)\|P(C|X, Y, Z)) - \mathrm{KL}(Q(C|X)\|P(C))\\
&\geq \mathbb{E}_{Q(C|X)}[\log P(Y|X, Z, C)] - \mathrm{KL}(Q(C|X)\|P(C))\\
&= \mathbb{E}_{Q(C|X)}[\log P(Y|X, Z, C)] + \mathbb{H}[Q(C|X)] + \log P(C).
\end{aligned}
\tag{2}
$$

In the second term of the third line of Eq.2, the distribution of environmental characteristics $Q(C|X)$ is adjusted toward the posterior probability $P(C|X, Y, Z)$. Since the latter is unknown, this term is omitted. Meanwhile, the third term $Q(C|X)$ ensures that the environmental encoding approximates the prior $P(C)$. This can also be interpreted as maximizing the entropy of the environmental encoding to capture more information while satisfying both prior and posterior probabilities. Consider realization, the environmental encoding $Q(C|X)$ can be derived using the reparameterization trick (Kingma, 2013) with a multivariate normal distribution to produce a d-dimensional contextual representation. The prior $P(C)$ serves as a key constraint for context generation. In models based on CVAE framework (Sohn et al., 2015; Xu et al., 2022a), it is typically set as a standard normal distribution. In the following sections, we will explore methods for obtaining a more optimal prior distribution. We refer to Eq.2 as $L_e$, representing the evidence lower bound (ELBO) for intervening confounding factors. This includes the prediction loss for future trajectories $P(Y|X, Z, C)$ and the previously mentioned KL divergence. According to the Markov property, $P(Y|X, Z, C) = P(Y|X, Z)P(Z|X, C)$. For $P(Z|X, C)$, the intermediate representation $P(Z|X) \sim N(\mu_1, \omega_1)$ is obtained using the trajectory encoder, while $Q(C|X) \sim N(\mu_2, \omega_2)$ serves

as the posterior distribution representing $C$. These distributions are combined to derive the optimal mixed distribution $P(Z|X,C) \sim N(\mu^*, \omega^*)$.

$$\mu^* = \frac{\omega_1^2 \mu_2 + \omega_2^2 \mu_1}{\omega_1^2 + \omega_2^2}, \quad \omega^* = \sqrt{\frac{\omega_1^2 \omega_2^2}{\omega_1^2 + \omega_2^2}}. \tag{3}$$

The decoder $P(Y|X,Z)$ is then used to predict future paths, as shown in Fig.2. Following previous studies, we also consider the distribution for reconstructing past trajectories $P(X|Z)$. The reconstruction loss is computed using the L2 norm between the reconstructed trajectory $X'$ and the observed $X$, denoted as $L_r = \|X' - X\|_2$.

# 3 PRIOR CONTINUAL LEARNING

While the proposed causal models bias by controlling for confounding factors, real-world scenarios often involve dynamic changes that can cause posterior estimates to deviate from optimal values or become stuck in local optima. Two key issues arise: first, catastrophic forgetting, where performance in earlier scenarios declines as new scene data is introduced; and second, the model's limited domain generalization, reflected in its inability to adapt to different scenarios.

In our framework, the prior distribution $P(C)$ represents the inherent nature of confounding factors within the environment, and high-quality prior significantly enhances the expressiveness of the estimator. Several prior studies have explored the selection of appropriate prior distributions. Research by Hoffman & Johnson (2016) and Tomczak & Welling (2018) demonstrates that using a simple gaussian prior can limit the expressiveness of variational inference. Studies such as Makhzani et al. (2015) and Tomczak & Welling (2018) propose using a posterior aggregation estimator to improve prior representation, while Egorov et al. (2021) introduces trainable pseudo-parameters optimized alongside the ELBO loss. Inspired by the aforementioned studies, we explore how to approximate optimal prior estimation using trajectory samples.

In the context of continuously changing tasks, the sample information under different environments can be represented as $P(E_i(X))$, while the corresponding contextual posterior is defined as $Q_i(C|X)$, the problem addressed by Eq.2 can correspondingly be written as:

$$(2) = \mathbb{E}_{Q(C|X)} \log P(Y|X,Z,C) - \sum_{i=1}^{K} \mathbb{E}_{\substack{P(E_i(X)) \\ Q_i(C|X)}} \mathrm{KL}[Q_i(C|X)\|\hat{P}(C)], \tag{4}$$

where $E_i$ represents the $i$-th environment in which the training set is located, the trajectory features' distribution changes continuously with environmental factors. $Q_i(C|X)$ represents the optimal posterior for each environmental factor, while $\hat{P}(C)$ represents the iteratively updated environmental prior. The optimization problem can be seen as iteratively updating a continuously changing prior that better adapts to the current environment. Assuming the initial prior $\hat{P}_1(C)$ is credible, when executing task $K$, the change in the second item of Eq.4 can be expressed as follows:

$$\sum_{i=1}^{K} \mathbb{E}_{P(E_i(X))} \mathrm{KL}[Q_i(C|X)\|\alpha_{K-1}(\dots \alpha_2(\alpha_1 \hat{P}_1(C) + (1-\alpha_1)\hat{P}_2(C)) \tag{5}$$
$$+ (1-\alpha_2)\hat{P}_3(C) + \dots) + (1-\alpha_{K-1})\hat{P}_K(C)]$$

For the convenience of calculation, We define $M_{\leq K-1}(C) = \prod_{j=1}^{K-1} \alpha_j \hat{P}_1(C) + \prod_{j=2}^{K-1} \alpha_j (1 - \alpha_1)\hat{P}_2(C) + \dots + \alpha_{K-1}(1 - \alpha_{K-2})\hat{P}_{K-1}(C)$. Inspired by Egorov et al. (2021), Eq.5 can be simplified as follows according to the calculation formula of limit:

$$(5) = \sum_{i=1}^{K} \mathbb{E}_{P(E_i(X))}[\mathrm{KL}[Q_i(C|X)\|M_{\leq K-1}(C)] - (1-\alpha_{K-1})(\underbrace{\hat{P}_K(C)\frac{Q_i(C|X)}{M_{\leq K-1}(C)} - 1})] + o(\alpha_{K-1}). \tag{6}$$

The detailed derivation process can be found in the App. A.2.2. Building on the convex property, we optimize two variables alternately: $\hat{P}_K(C)$, representing the shift in the prior probability density function, and $\alpha_k$, which adjusts the scaling factor. For step $K$, the first term in Eq.6 is fixed, while the optimal value of $\hat{P}_K(C)$ is derived from the target posteriors $Q_i(C|X)$ of the currently observed samples and the obtained prior $M_{\leq K-1}(C)$ from previous steps.

In line with continual learning theory, we design a prior queue to store a representative set for each task. The environment-specific prior align with the trajectory feature fingerprints stored in memory to optimize task performance. A straightforward approach uses the posterior from inputs to approximate the optimal prior for different environments, constrained by $\int P(C)\mathrm{d}C = 1$. As noted by Tomczak & Welling (2018), relying solely on posterior probability risks of overfitting and entirely storing all data is impractical. Instead, we employ multiple pseudo-features $U$ to approximate $P(E_i(X))$ for the corresponding scenario.

$$\hat{P}_i(C) = \frac{1}{|\mathcal{M}_i|} \sum_{U_i \in \mathcal{M}_i} Q(C|U_i), \tag{7}$$

where $\mathcal{M}_i$ is the prior set related to the $i$-th scene stored in the memory queue. Using the online or offline mechanisms discussed later, we obtain prior components closely associated with the scenarios. These components are iteratively optimized with scene data during training and stored in a dedicated prior queue.

## 4 TRAINING PROCESS

To streamline the training process, we design a min-max method that promotes iterative optimization. In the minimization step, the goal is to update the prior $\hat{P}_K(C)$ for the current task, as in Eq.6 and Eq.7, by solving for the corresponding pseudo feature $U_K$. Recalling the optimization objective of the probability density function, we aim to minimize Eq.6, where the first term is a constant value that can be ignored, and we emphasize the target probability to be optimized with an underline. Concurrently, to improve the robustness of the representation process, we introduce an information entropy loss for $\hat{P}_K(C)$, resulting in the following:

$$
\begin{aligned}
\hat{P}_K^*(C) &= \arg\min(\mathbb{E}(\hat{P}_K(C)\log(\hat{P}_K(C)) - \log(\hat{P}_K(C)\frac{\sum_{i=1}^{K} Q_i(C|X)}{M_{\leq K-1}(C)}))) \\
&\leq \arg\min(\mathbb{E}(\hat{P}_K(C)\log(\hat{P}_K(C)) - \hat{P}_K(C)\log(\frac{\sum_{i=1}^{K} Q_i(C|X)}{M_{\leq K-1}(C)}))) \\
&= \arg\min \mathrm{KL}[\hat{P}_K(C)\|\frac{\sum_{i=1}^{K} Q_i(C|X)}{M_{\leq K-1}(C)}].
\end{aligned}
\tag{8}
$$

Where the second line is derived based on Jensen's inequality. The optimized prior $\hat{P}_K^*(C) = Q(C|U_K)$ is computed from the trainable pseudo feature set $U_K$ of the current task $K$, while $M_{\leq K-1}(C)$ represents the optimal prior maintained up to round $K-1$. The posterior probability of the environmental variable $Q_i(C|X)$ is the target to be solved and can be approximately expressed as $\frac{|D^{\leq K-1}|}{|D^{\leq K}|}Q(C|U^{\leq K-1}) + \frac{|D^K|}{|D^{\leq K}||T|} \sum_{T \sim D_K} Q(C|X_T)$. Given the temporal nature of trajectory data, we design both offline and online update mechanisms to generate pseudo features $U$ to the memory queue.

**Online Mode:** The online mode is selected when access to the full training dataset is limited, or during streaming processes where historical data cannot be stored, such as when adapting autonomous vehicles to new regions. In each task $K$, one pseudo feature $U_K^i$ is incrementally added in a streaming manner to solve for the prior. However, unlike structured data such as images that can be processed randomly, trajectory data requires careful handling due to its temporal inductive bias. We initialize $U_K^i$ using agents' trajectories $X^i$ from the current batch $B_K$, where divergence is optimal, determined by $\underset{X^i \in B_K}{\arg\min} \mathrm{KL}(Q(C|X^i)\|\sum_{i=1}^{K} Q_i(C|X))$. To prevent overfitting, we introduce trainable random noise $\epsilon \sim N(1,0)$ for the initialization. Once the optimal prior is obtained, its corresponding weights must be close to the shape of the aggregated posterior distribution, as specified by the first line of Eq.6:

$$\alpha_{K-1}^* = \mathrm{KL}[\sum_{i=1}^{K} Q_i(C|X)\|(\alpha_{K-1}M_{\leq K-1}(C) + (1 - \alpha_{K-1})\hat{P}_K(C))]. \tag{9}$$

**Offline Mode:** In offline mode, where all training data is available in advance, selecting the highest-quality fingerprint vectors for the current task helps avoid local optima. We designed a simple clustering-based method, similar to Wang et al. (2023), employing an LSTM+CNN approach to extract pre-trained feature values. The cluster centers serve as the informative pseudo-features for the

current task, and trainable noise $\epsilon \sim N(1,0)$ is uniformly added to them. Unlike the online mode, pseudo-features are added in batches at the beginning of the task, and all samples and corresponding weights are updated in each iteration, with loss functions similar to Eq.8 and Eq.9.

The optimization is continually refined with each training batch iteration using the formula above. In the maximization step, we optimize according to Eq.2, with a focus on the KL divergence constraint term. To reduce optimization difficulty while simultaneously improve the training efficiency, we use a pre-trained trajectory encoder as the initial parameter for $Q(C|X)$. To enhance the diversity of pseudo-features and enable them to capture more information, we adopt a regularization approach used in Egorov et al. (2021) which increases the symmetric KL divergence between $Q(C|U_K)$ from the current iteration and the mixture component latent distribution $M_{\leq K-1}(C)$ from the previous iteration. The pseudo-labels are better aligned to extract information of the current iteration under this constraint. Based on all of the above analysis, the loss function for the maximization step is formulated as follows:

$$\max \mathbb{E}_{\substack{P(E_K(X)) \\ Q(C|X)}} [P(Y|X,Z,C) + P(X|Z) - \mathrm{KL}[Q(C|X)\|\hat{P}_K(C)]] + \mathrm{KL}_{\mathrm{sym}}[Q(C|U_K)\|M_{\leq K-1}(C)].$$
(10)

**Pruning:** For generalization across different regions, it is essential to maintain component expansion within a controllable range while preserving maximal diversity. A common method involves using clustering techniques to identify similar prior clusters, followed by pruning. However, given the sparsity of the distribution, obtaining meaningful clusters can be challenging. Drawing inspiration from Ye & Bors (2023), we propose a pruning strategy that introduces greater diversity into the priors across multiple tasks. Specifically, if two pseudo-features encapsulate identical critical information, their corresponding latent variables are expected to be similar. We calculate the similarity $S(\cdot, \cdot)$ between each pair of $\{Q(C|U^i)\}_{i=1}^{|U|}$ and identify the closest pair of variables, deeming them redundant:

$$a^*, b^* = \arg \min_{\substack{a \neq b \\ a,b \in [1,|U|]}} S(a,b).$$
(11)

We evaluate $S(a,b)$ by calculating the squared loss $\| \cdot \|_2$. The indices $a^*$ and $b^*$ represent the selected candidate pairs. We then compute the differences between these two representations and the remaining pseudo-features, pruning the feature with the smallest difference.

$$I^* = \mathrm{argmin}\{ \sum_{\substack{j=1 \\ j \neq a^*, b^*}}^{|U|} S(a^*, j), \sum_{\substack{j=1 \\ j \neq a^*, b^*}}^{|U|} S(b^*, j)\}.$$
(12)

The index $I^*$ represents the selected index for pruning. We iteratively perform the pruning operation to ensure that the number of features remains below the threshold $\gamma$.

This training mechanism is plug-and-play, accommodating generic encoders and decoders. Prediction can be made directly using the original model during the inference phase. The detailed process of our proposed model can be referred to Algorithm1 in App. A.1.

## 5 EXPERIMENTS

We validate the superiority of the proposed multi-agent trajectory prediction model, $C^2$INet, using both real-world datasets encompassing multiple scenarios and synthetic datasets. The model's performance is validated using two widely adopted trajectory prediction metrics: Average Displacement Error (ADE) and Final Displacement Error (FDE). ADE measures the mean squared error (MSE) between predicted and ground-truth trajectories, while FDE calculates the L2 distance between their final positions. Following prior studies, we generate 20 samples from the predicted distribution and select the one closest to the ground truth for both metrics. Each metric is evaluated five times using different random seeds, and the average value is reported. Due to space limitations, some experimental analysis conclusions can be found in App. A.4.

### 5.1 DATASETS

**ETH-UCY Dataset:** This widely used dataset contains the trajectories of 1,536 pedestrians in real-world scenarios, capturing complex social interactions (Lerner et al., 2007). The data was collected at 2.5 Hz (one sample every 0.4 seconds). Following the experimental setups of Liu et al. (2022); Chen et al. (2021a); Bagi et al. (2023), we use 3.2 seconds (8 frames) of observed trajectories to predict the next 4.8 seconds (12 frames). The dataset includes five distinct environments:

{hotel, eth, univ, zara1, zara2}. Each task is trained for 300 epochs, corresponding to the unique environments in our continual learning framework.

**Synthetic Dataset:** This dataset is used to evaluate the effectiveness of continual learning for trajectories across different motion styles. It is based on the framework from Liu et al. (2022), capturing multi-agent trajectories in circle-crossing scenarios. The dataset includes various minimum distance settings between pedestrians: $\{0.1, 0.2, 0.3, 0.4, 0.5, 0.6, 0.7, 0.8\}$ meters. Each domain consists of 10,000 training trajectories, 3,000 validation trajectories, and 5,000 test trajectories. The model uses 8-frame observation segments to predict future 12-frame trajectories.

**SDD Dataset:** The Stanford Drone Dataset (SDD) (Robicquet et al., 2016) is a large-scale collection of images and videos featuring various agents across eight distinct scene regions, making it well-suited for continual learning setups. The dataset captures over 40,000 interactions between agents and the environment and more than 185,000 interactions between agents, providing a rich foundation for trajectory prediction tasks in highly interactive scenarios.

## 5.2 BASELINES

We select specific baselines for the continual training scenarios discussed in this paper. To assess the generality of our counterfactual analysis method, we integrate it as a plug-and-play module into two baseline models: the RNN-based STGAT (Alahi et al., 2016) and the CNN-based SocialSTGCNN (Zhao et al., 2019). We also experiment with the CVAE-based PECNet (Mangalam et al., 2020) and YNet (Mangalam et al., 2021) which incorporates scene map information from the SDD dataset, both based on an encoder-decoder architecture. Additionally, we compare other causal intervention methods, such as COUNTERFACTUAL (Chen et al., 2021a) and INVARIANT (Liu et al., 2022) (reporting the best-performing $\mu = 1.0$). We select GCRL (Bagi et al., 2023) and EXPANDING (Ivanovic et al., 2023) for continual learning methods, with the latter implemented by us as it is not open-source. These methods are based on domain adaptation, where the prior distribution is retrained for new scenarios without using previous data. We also compare our approach with typical continual learning methods (using STGAT as backbone), including Elastic Weight Consolidation (EWC) (Kirkpatrick et al., 2017), which applies gradient constraints; latent features modeled as a mixture of Gaussians with diagonal covariance (MoG); and the random coresets method (Coresets) (Bachem et al., 2015), which uses memory replay during training. Our analysis shows that our approach effectively captures high-quality changes in environmental content.

## 5.3 RESULTS IN CONTINUAL LEARNING SETTING

Table1 below and Table2-3 in App. A.4.1 present comprehensive comparisons of the average performance for current and previously completed tasks under continual learning, evaluated across three datasets categorized by task scenarios. In the first column, a colon preceding the task name denotes the combination of the current and completed task sample sets. We test the proposed $C^2$INet in both online and offline modes, employing STGAT and STGCNN as backbones to compare $C^2$INet's performance with that of GCRL.

On the ETH-UCY dataset, $C^2$INet-offline consistently delivers the best performance across the first four tasks. In contrast, $C^2$INet-online shows slightly weaker results, mainly due to the challenges of acquiring high-quality prior in the online mode, which increases the risk of the model converging to local optima. Compared to the original STGAT and STGCNN models, continual causal intervention provides at least a $10.4\%$ improvement, particularly as tasks are progressively added. This demonstrates that our method significantly enhances model stability at a low computational cost, with the Continual Memory module substantially boosting the model's generalization capacity. For alternative causal intervention methods like COUNTERFACTUAL, the debiasing effect is most evident in complex environments, such as the "hotel" scenario, where it ultimately achieves superior results. However, traditional continual learning methods such as EWC and random Coresets cannot outperform our approach, lagging by at least $2.5\%$ on metrics like ADE.

On the Synthesis dataset containing fixed noise, explicit causal intervention methods like COUNTERFACTUAL exhibited more substantial advantages. Nonetheless, both versions of $C^2$INet remained competitive. Specifically, $C^2$INet-online with STGAT demonstrate a slight edge in the ":0.3" and ":0.4" tasks, while $C^2$INet-offline outperform in the ":0.2", ":0.5", and ":0.6" tasks. Overall, the relatively stable Synthesis dataset supports strong performance across various continual learning strategies. In the "0.1" task, GCRL paired with STGAT initially performs well, but as the number

Table 1: The table presents the model's average performance across all previously encountered tasks on ETH-UCY dataset. The first column lists the newly added tasks in the continual learning setup, with the preceding colon indicating the average trajectory prediction results for all tasks encountered up to that point. All results are averaged over five runs, with the best outcomes highlighted in bold and the second-best results underlined. Color blocks indicate the ranking of different backbones for comparison.

| TASK | univ | :eth | :zara1 | :zara2 | :hotel |
|---|---|---|---|---|---|
| STGAT | 0.67/1.35 | 0.96/1.92 | 0.84/1.66 | 0.63/1.26 | 0.96/1.92 |
| STGCNN | 0.82/1.66 | 1.27/2.37 | 1.05/2.04 | 0.76/1.52 | 1.33/2.55 |
| PECNet | 0.60/1.27 | 0.89/1.85 | 0.71/1.51 | 0.60/1.27 | 0.91/1.83 |
| COUNTERFACTUAL | 0.58/1.21 | 0.86/1.82 | 0.67/1.40 | 0.52/1.12 | **0.83/1.75** |
| INVARIANT | 0.61/1.35 | 1.05/2.21 | 0.76/1.70 | 0.69/1.56 | 1.07/2.06 |
| ADAPTIVE | 0.87/1.69 | 1.13/2.19 | 0.84/1.81 | 0.61/1.38 | 0.99/1.96 |
| EWC | 0.57/1.23 | 0.84/1.73 | 0.71/1.41 | 0.59/1.29 | 0.98/1.89 |
| MoG | 0.68/1.35 | 0.84/1.81 | 0.77/1.46 | 0.69/1.31 | 1.01/1.98 |
| Coresets | 0.59/1.25 | 0.89/1.87 | 0.57/1.26 | 0.52/1.14 | 0.95/1.89 |
| $\text{GCRL}_{+STGAT}$ | 0.79/1.59 | 1.12/2.21 | 0.71/1.48 | 0.53/1.13 | 0.94/2.01 |
| $\text{GCRL}_{+STGCNN}$ | 0.82/1.58 | 1.24/2.31 | 0.99/1.93 | 0.69/1.38 | 1.27/2.54 |
| $\text{C}^2\text{INet-online}_{+STGAT}$ | 0.52/1.13 | 0.79/1.65 | 0.54/1.19 | 0.49/1.10 | 0.86/1.82 |
| $\text{C}^2\text{INet-online}_{+STGCNN}$ | 0.64/1.32 | 0.94/1.89 | 0.72/1.49 | 0.59/1.27 | 0.98/1.95 |
| $\text{C}^2\text{INet-offline}_{+STGAT}$ | **0.51/1.09** | **0.75/1.58** | **0.54/1.14** | **0.48/1.03** | 0.86/1.81 |
| $\text{C}^2\text{INet-offline}_{+STGCNN}$ | 0.57/1.19 | 0.91/1.87 | 0.69/1.42 | 0.56/1.19 | 0.95/1.91 |

of tasks increases, its performance degrades due to catastrophic forgetting, ultimately falling behind our proposed method.

On the SDD dataset, YNet, which incorporates map information to support trajectory prediction, excels in several tasks. The performance gap between STGAT and SocialSTGCNN is minimal; however, their corresponding $\text{C}^2\text{INet}$ variants produce performance improvements of around $1.5\%$ to $7\%$ across different tasks. COUNTERFACTUAL performs optimally during the early stages of training, underscoring its strong debiasing effect. However, as tasks accumulate, the forgetting effect allows $\text{C}^2\text{INet-offline}$ to take the lead gradually. In this dataset, the offline mode outperforms the online mode, which can be attributed to the complexity of SDD tasks and the diversity of agent types. The offline model demonstrates better resilience in avoiding local optima under these conditions.

In conclusion, our proposed model exhibits robustness across multiple datasets, consistently enhancing trajectory prediction performance and effectively mitigating the challenge of catastrophic forgetting. We also provide detailed metric analysis for each task, which is illustrated in App. A.4.3.

## 5.4 TRAINING PROCESS ANALYSIS

Fig.4 in App. A.4.2 illustrates the ADE's statistics for each model across different tasks during the training process. It shows that performance on previously completed tasks deteriorates with the introduction of new tasks. The performance curves for STGAT and GCRL follow similar trends, exhibiting overall stability. Specifically, for the STGAT model, performance on the "univ" task declines sharply after adding the "eth" task. At the same time, GCRL experiences a more gradual performance decline following introducing the "hotel" task, indicating a greater resilience to optimization loss. In contrast, STGAT displays more abrupt, stepwise performance shifts.

The EWC and COUNTERFACTUAL exhibit greater volatility in performance throughout training, particularly on the "eth", "zara1" and "zara2" tasks. COUNTERFACTUAL, in particular, shows sharp fluctuations, resulting in significant variations in its average performance. By comparison, $\text{C}^2\text{INet-offline}$ consistently delivers superior results, with its Continual Memory module effectively retaining information from prior tasks, thereby guiding the optimization process and enabling stable performance improvements across all tasks. While $\text{C}^2\text{INet-online}$ performs slightly below its offline counterpart, its integrated training strategy produces smoother performance curves, reflecting more excellent stability throughout the training process.

## 5.5 QUALITATIVE RESULTS

Fig.8 in App. A.4.4 visually represents the prediction results across multiple scenarios within the ETH-UCY dataset, utilizing randomly selected samples under consistent training configurations as previously described. The gray lines depict the observed trajectories, the red points indicate the predicted paths and the green points represent the ground truth. From the experimental outcomes, several key conclusions can be drawn: (1) Both $C^2$INet-online and $C^2$INet-offline exhibit robust predictive performance across the tested scenarios. (2) In contrast, the GCRL and STGAT models demonstrate noticeable performance degradation, primarily due to the residual effects of prior training tasks. For example, errors such as incorrectly predicting a straight path as a turn (eth #42, #43) or misclassifying a right turn as a left turn (eth #37, #39) are evident.

Overall, our proposed method significantly enhances the stability and reliability of trajectory prediction models by integrating the Causal Intervention and Continual Memory mechanisms. This improvement is particularly pronounced in scenarios where slight changes in direction are often misclassified as sharp turns—an issue commonly observed in STGAT and GCRL models. Such errors are primarily attributable to noise introduced in multi-task environments, which impairs the models' predictive accuracy. By effectively addressing these challenges, $C^2$INet offers a more resilient approach to handling complex trajectory prediction tasks.

## 5.6 VISUALIZATION ANALYSIS

In this section, we present a visualization of the two-dimensional posterior probability $Q(C|X)$ across various tasks to provide insights into the model's performance improvements. As depicted in Fig.3, the priors derived from task-specific memories encode highly distinct posterior distributions. Following training on five separate tasks from the ETH-UCY dataset, a well-defined mixture of Gaussian distributions emerges in both $C^2$INet-online and $C^2$INet-offline. This result can be attributed to the imposed prior constraints, which allow the model to effectively differentiate between new tasks while retaining knowledge of previously learned ones. In contrast, the variant posterior $Q(S|X)$ produced by GCRL on both backbones (STGAT and STGCNN) exhibits an indistinguishable, aggregated distribution. This lack of discriminative task representation leads to confusion between tasks during continual updates, highlighting the limitations of GCRL in preserving task-specific knowledge. The observed differences explain the importance of distinct posterior distributions in enhancing model adaptability and preventing task interference in continual learning scenarios.

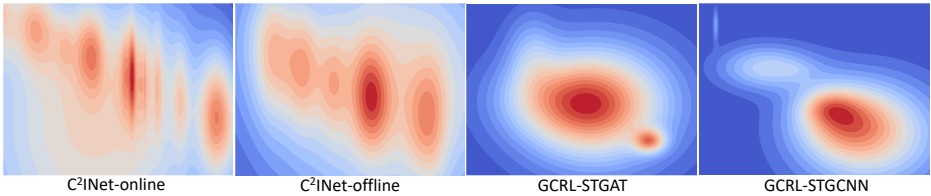

Figure 3: Visualization of the posterior distribution for different models with 2D latent space.

## 6 CONCLUSION

In this paper, we focus on developing a trajectory data mining method that can generalize across different domains through end-to-end training, even under continual learning conditions where data from all task scenarios cannot be accessed simultaneously. First, we analyze the spurious correlations introduced by complex real-life environments and propose a plug-and-play Continual Causal Intervention ($C^2$INet) framework that mitigates confounding factors affecting representations. Considering the practical challenges of incomplete data collection or limited device resources, we innovatively combine the proposed trajectory learning framework with a continual learning strategy. Experiments on multiple types of datasets fully demonstrate that the proposed $C^2$INet model effectively addresses the issue of catastrophic forgetting and enhances the robustness of trajectory prediction.

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

# A   APPENDIX

## A.1   TRAINING PROCESS

The following Alg.1 provides a detailed description of the training process for the proposed $C^2$INet method.

---

**Algorithm 1** Training Process of $C^2$INet.

---

**Input:** The training trajectories from $K$ task domains $\{X_i, Y_i\}_{i=1}^K$, with a maximum prior queue capacity of $\gamma$, training $L$ epochs for each task.
**Output:** The optimized model parameters $\theta^*$.
1: **for** each i $\in [1, K]$ **do**
2:   **for** each j $\in [1, L]$ **do**
3:     **if** j $= 1$ **then**
4:       **if** *Online Mode* **then**
5:         Sample $S$ trajectories from the current task.
6:         Obtain the initial prior based on the sampled data and enqueue it.
7:       **else if** *Offline Mode* **then**
8:         Cluster the accessible data and enqueue the cluster centers.
9:       **end if**
10:     **end if**
11:     Maximize the loss function of the causal intervention model Eq.10 to optimize model parameters $\theta$.
12:     **if** j $\mod \lfloor \frac{L}{\gamma} \rfloor = 0$ **then**
13:       **if** *Online Mode* **then**
14:         Calculate the new component based on Eq.8 and Eq.9 and add it to the prior queue.
15:       **else if** *Offline Mode* **then**
16:         Optimize the obtained components in the prior queue based on Eq.8 and Eq.9.
17:       **end if**
18:     **end if**
19:   **end for**
20:   **if** the queue length exceeds $\gamma$ **then**
21:     Pruning is performed according to Eq.10 until the quantity is reduced below $\gamma$.
22:   **end if**
23: **end for**
24: **return** The optimized model parameters $\theta^*$

---

## A.2   DETAILS SUPPLEMENT

### A.2.1   DERIVATIONS OF THE CAUSAL INTERVENTION IN SECTION 2.3

We begin by introducing causal interventions and the related do-calculus techniques, which can be regarded as the axioms underlying our methodological derivations. Let us assume $P(Y|X)$ represents the conditional probability of $Y$ given the variable $X$ as an observed known. Clearly, within the system, if other confounding factors are informationally associated with $x$, then $P(Y|X)$ cannot accurately measure the direct relationship from $X$ to $Y$. To address this issue, the formula $P(Y|do(X))$ is used to represent an intervention applied to $X$, specifically assigning a fixed value $X = x$, severing the informational pathways from other variables to $X$, thereby obtaining the direct effect between the variables. The identification of causal effects here follows the backdoor criterion (Pearl, 2016). As we will see, the do-calculus provides us with tools to identify causal effects using the causal assumptions encoded in the causal graph. It consists of three inference rules that allow us to map interventional and observational distributions whenever certain conditions are satisfied in the Structural Causal Models $G$, which is a Directed Acyclic Graph (DAG) that describes causal attributes and their interactions (Neal, 2020). Let $X, Y, Z$, and $W$ be arbitrary disjoint sets of nodes in a causal DAG $G$. Let $G_{\overline{X}}$ denote the graph obtained by deleting from $G$ all arrows pointing to nodes in $X$ and $G_{\underline{X}}$ denote the graph obtained by deleting from $G$ all arrows emerging from nodes

in $X$. To represent the deletion of both incoming and outgoing arrows, we use the notation $G_{\overline{X}\underline{Z}}$. The following three rules are valid for every interventional distribution compatible with $G$.

**Rule 1** (Insertion/deletion of observations):

$$P(y|do(x), z, w) = P(y|do(x), w), \text{if}(Y \perp\!\!\!\perp Z|X, W)_{G_{\overline{X}}}. \tag{13}$$

**Rule 2** (Action/observation exchange):

$$P(y|do(x), do(z), w) = P(y|do(x), z, w), \text{if}(Y \perp\!\!\!\perp Z|X, W)_{G_{\overline{X}\underline{Z}}}. \tag{14}$$

**Rule 3** (Insertion/deletion of actions):

$$P(y|do(x), do(z), w) = P(y|do(x), w), \text{if}(Y \perp\!\!\!\perp Z|X, W)_{G_{\overline{X}\overline{Z(W)}}}. \tag{15}$$

Next, we present the derivation process for $P(Y|do(X))$ in the main text. Specifically, based on the intervention on $X$ in the Structural Causal Model as depicted in Fig. 2 and the formula for conditional probability, we can deduce:

$$
\begin{aligned}
P(Y|do(X)) &= \mathbb{E}_{\substack{P(Z)\\P(C)}} P(Y|do(X), Z, C)\\
&= \mathbb{E}_{\substack{P(Z)\\P(C)}} P(Y|do(X), Z, C)P(Z|do(X), C)P(C|do(X))\\
&= \mathbb{E}_{\substack{P(Z)\\P(C)}} P(Y|X, Z, C)P(Z|X, C)P(C|X).
\end{aligned}
\tag{16}
$$

In the above process, given that $Y \perp\!\!\!\perp X|Z, C$ and $Z \perp\!\!\!\perp X|C$ in $G_{\underline{X}}$, according to the *Action/observation exchange* Rule, it can be derived that $P(Y|do(X), Z, C) = P(Y|X, Z, C)$ and $P(Z|do(X), C) = P(Z|X, C)$. Similarly, with $C \perp\!\!\!\perp X$ in $G_{\overline{X}}$, by applying the *Insertion/Deletion of Actions* Rule, it can be deduced that $P(C|do(X)) = P(C|X)$.

### A.2.2 Derivations Of The Objective Function In Section 3

Eq. 4 presents our optimization objective, with the detailed derivation of the second term, the KL divergence, as follows:

$$\sum_{i=1}^{K} \mathbb{E}_{\substack{P(E_i(X))\\Q_i(C|X)}} \text{KL}[Q_i(C|X)\|\hat{P}(C)]$$

$$= \sum_{i=1}^{K} \mathbb{E}_{P(E_i(X))} \text{KL}[Q_i(C|X)\|\alpha_{K-1}(\ldots\alpha_2(\alpha_1\hat{P}_1(C) + (1-\alpha_1)\hat{P}_2(C))$$

$$+ (1-\alpha_2)\hat{P}_3(C) + \ldots) + (1-\alpha_{K-1})\hat{P}_K(C)]$$

$$= \sum_{i=1}^{K} \mathbb{E}_{P(E_i(X))} \text{KL}[Q_i(C|X)\| \prod_{j=1}^{K-1} \alpha_j\hat{P}_1(C) + \prod_{j=2}^{K-1} \alpha_j(1-\alpha_1)\hat{P}_2(C) + \cdots + (1-\alpha_{K-1})\hat{P}_K(C)].$$

$$\tag{17}$$

For the convenience of calculation, We define $M_{\leq K-1}(C) = \prod\limits_{j=1}^{K-1} \alpha_j\hat{P}_1(C) + \prod\limits_{j=2}^{K-1} \alpha_j(1-\alpha_1)\hat{P}_2(C) + \cdots + \alpha_{K-1}(1-\alpha_{K-2})\hat{P}_{K-1}(C)$. Inspired by Egorov et al. (2021), Eq.17 can be

simplified as follows according to the calculation formula of limit:

$$(17) = \sum_{i=1}^{K} \mathbb{E}_{P(E_i(X))} \mathrm{KL}[Q_i(C|X) \| (\alpha_{K-1} M_{\leq K-1}(C) + (1 - \alpha_{K-1})\hat{P}_K(C))]$$

$$= \sum_{i=1}^{K} \mathbb{E}_{P(E_i(X))} \mathbb{E}_{Q_i(C|X)} [\log \frac{Q_i(C|X)}{\alpha_{K-1} M_{\leq K-1}(C) + (1 - \alpha_{K-1})\hat{P}_K(C)}]$$

$$= -\sum_{i=1}^{K} \mathbb{E}_{P(E_i(X))} \mathbb{E}_{Q_i(C|X)} [\alpha_{K-1} \log \frac{M_{\leq K-1}(C)}{Q_i(C|X)} + \log(1 + \frac{(1 - \alpha_{K-1})\hat{P}_K(C)}{\alpha_{K-1} M_{\leq K-1}(C)})]$$

$$= -\sum_{i=1}^{K} \mathbb{E}_{P(E_i(X))} \mathbb{E}_{Q_i(C|X)} [\log \frac{M_{\leq K-1}(C)}{Q_i(C|X)} + \log(\alpha_{K-1} + \frac{(1 - \alpha_{K-1})\hat{P}_K(C)}{M_{\leq K-1}(C)})]$$

$$= -\sum_{i=1}^{K} \mathbb{E}_{P(E_i(X))} \mathbb{E}_{Q_i(C|X)} [\log \frac{M_{\leq K-1}(C)}{Q_i(C|X)} + \log((1 - \alpha_{K-1})(\frac{\hat{P}_K(C)}{M_{\leq K-1}(C)} - 1) + 1)]$$

$$\approx \sum_{i=1}^{K} \mathbb{E}_{P(E_i(X))} [\mathrm{KL}[Q_i(C|X) \| M_{\leq K-1}(C)] - (1 - \alpha_{K-1})(\hat{P}_K(C) \underbrace{\frac{Q_i(C|X)}{M_{\leq K-1}(C)}} - 1)] + o(\alpha_{K-1}).$$

$$(18)$$

The derivation of the penultimate line utilizes an approximation by taking the limit value.

## A.3 RELATED WORK

### A.3.1 LEARNING-BASED TRAJECTORY PREDICTION

End-to-end trajectory prediction using deep learning has become the mainstream because it can account for multiple factors contributing to prediction accuracy. The most common approach involves sequential networks, which take the past path points of multiple agents across several frames, along with various attributes, to predict future movements. These networks include Recurrent Neural Networks (RNN) (Zyner et al., 2018; 2017), Long Short-Term Memory (LSTM) models (Alahi et al., 2016; Salzmann et al., 2020; Xin et al., 2018), and Graph Convolutional Networks (GCN) (Shi et al., 2021; Li et al., 2019). These architectures extract attributes like speed, direction, road characteristics, and interactions, yielding high-dimensional vector representations or feature distributions in latent space. Decoders or sampling methods then use these representations to predict future trajectories.

Attention-based methods extract relational patterns in both time and space (Wu et al., 2021; Kim et al., 2020; Messaoud et al., 2020), with attention mechanisms controlling the flow of critical information. Transformer-based architectures are prominent in this category. For instance, Liu et al. (Liu et al., 2021) propose a multi-modal architecture with stacked transformers to capture features from trajectories, road data, and social interactions. Similarly, Zhao et al. Zhao et al. (2021) utilize a transformer with residual layers and pooling operations to integrate geographical data for learning interactions. The Spatio-Temporal Transformer Networks (S2TNet) (Chen et al., 2021b) use a spatio-temporal transformer for interactions and a temporal transformer for sequences. Generative Adversarial Networks (GANs) (Gupta et al., 2018) also capture the data distribution, generating diverse and plausible trajectory predictions. Alternatively, large language models have been applied to generate trajectories based on semantics (Lan et al., 2024; Peng et al., 2024).

Recent works further integrate map or scene information for more comprehensive prediction. CNNs have been used to extract features from Bird's Eye View (BEV) representations (Mangalam et al., 2021; Chou et al., 2020), while context rasterization techniques address Vulnerable Road User (VRU) trajectory prediction (Cui et al., 2019; Djuric et al., 2020). Additionally, some works (Gu et al., 2022; Li et al., 2023; Bae et al., 2024; Li et al., 2024; Xu & Fu, 2024) that integrate emerging research methods such as diffusion models have been proposed, which can effectively enhance the performance of trajectory prediction models in challenging situations. This paper proposes a general training method for end-to-end trajectory prediction models that enhances generalization performance with plug-and-play flexibility.

### A.3.2 Causal Inference

The core objective of causal inference is to identify critical factors influencing outcomes. Structural causal models are commonly used for modeling, with principles like backdoor adjustment employed to intervene in causal relationships. In deep learning applications, the focus is often on analyzing confounding factors to mitigate the impact of perturbations on model performance, as demonstrated by methods in Johansson et al. (2016) and Wu et al. (2023). In recent years, causal intervention methods have been applied in trajectory prediction to enhance performance across domains. For example, Chen et al. (2021a) employs counterfactual interventions, such as using a zero vector, to reduce bias between training and deployment environments. Liu et al. (2022) highlights that target trajectory $Y$ is often correlated with observation noise and agent densities, proposing a gradient norm penalty over empirical risk to mitigate environmental effects (Bagi et al., 2023). Additionally, Pourkeshavarz et al. (2024) leverages disentangled representation learning to isolate invariant and variant features, minimizing the latter's influence on trajectory prediction. Our work addresses catastrophic forgetting in domain-shift scenarios by learning the prior distribution of trajectory representations across contexts to identify scenario-specific confounding factors. This approach enables intuitive manipulation through controlled interventions and adapts seamlessly to continuously evolving conditions.

### A.3.3 Continual Learning

Continual learning aims to maintain strong model performance as new domain samples are introduced, addressing catastrophic forgetting by balancing the retention of previous knowledge with the acquisition of new tasks. The most common approach uses constraint-based methods on the loss function, ensuring past learning directions are considered during gradient updates while adapting to new samples (Kirkpatrick et al., 2017; Lopez-Paz & Ranzato, 2017). However, these methods often lack interpretability and accuracy, especially when current tasks differ significantly from prior ones. An alternative approach is the rehearsal method (Wang et al., 2019; Riemer et al., 2018), which reinforces past knowledge by replaying previous samples during training. Memory-based mechanisms also preserve past information by storing samples as tasks accumulate (Chaudhry et al., 2019). Despite some work on expanding domain generalization through meta-learning (Ivanovic et al., 2023), limited research addresses continual learning in trajectory prediction, particularly in mitigating forgetting and adapting to changing scenarios. Our method integrates causal inference with a memory-based continual learning framework to develop a generalized trajectory prediction model.

### A.4 More about Experiments

#### A.4.1 Additional Results in Continual Learning Setting

Table2 and Table3 present the training results of continual learning on the Synthetic Dataset and SDD Dataset, corresponding to the results described in Section 5.3.

#### A.4.2 Training Process Curve

Fig.4 displays the ADE's statistics for each model across various tasks during the training process, corresponding to the results described in Section 5.4.

#### A.4.3 Metrics for each task separately

In Fig.5-7, we provide a detailed representation of performance metrics (ADE) across various tasks, all within the same continual learning framework outlined in Sec. 5.3. The severity of catastrophic forgetting differs across tasks for the ETH-UCY dataset, as depicted in Fig.5. Taking the "univ" task as an example, while most models maintain reasonable performance, the original STGAT shows the weakest results. In contrast, $C^2$INet performs significantly better due to its effective intervention in counteracting environmental influences on trajectory representations. When training progresses to the "eth" task, the task complexity increases significantly, resulting in an ADE of over 1.3 for all models. Additionally, the performance on the previously completed "univ" task deteriorates, with its ADE rising above 0.7. $C^2$INet exhibits the least amount of forgetting, as it effectively retains knowledge from past tasks. The fifth plot illustrates the effects of forgetting across tasks after completing the entire training cycle. It is evident that while the "hotel" task demonstrates post-training solid performance, it has a detrimental impact on other tasks, likely due to gradient

Table 2: The table presents the model's average performance across all previously encountered tasks on Synthetic Dataset. The first column lists the newly added tasks in the continual learning setup, with the preceding colon indicating the average trajectory prediction results for all tasks encountered up to that point. All results are averaged over five runs, with the best outcomes highlighted in bold and the second-best results underlined. Color blocks indicate the ranking of different backbones for comparison.

| TASK | 0.1 | :0.2 | :0.3 | :0.4 | :0.5 | :0.6 |
|---|---|---|---|---|---|---|
| STGAT | 0.15/0.20 | 0.17/0.24 | 0.19/0.25 | 0.21/0.31 | 0.25/0.33 | 0.28/0.36 |
| STGCNN | 0.65/1.16 | 0.50/0.92 | 0.39/0.75 | 0.36/0.62 | 0.39/0.67 | 0.42/0.65 |
| PECNet | 0.10/0.15 | 0.14/0.17 | 0.17/0.20 | 0.18/0.23 | 0.23/0.25 | 0.24/0.26 |
| COUNTERFACTUAL | 0.07/0.13 | **0.06/0.09** | **0.08/0.12** | **0.10/0.12** | **0.15/0.16** | **0.18/0.22** |
| INVARIANT | 0.07/0.15 | 0.11/0.16 | 0.15/0.18 | 0.21/0.19 | 0.28/0.24 | 0.37/0.38 |
| ADAPTIVE | 0.15/0.21 | 0.17/0.23 | 0.19/0.29 | 0.18/0.27 | 0.23/0.31 | 0.29/0.36 |
| EWC | 0.09/0.14 | 0.08/0.15 | 0.11/0.19 | 0.15/0.25 | 0.17/0.30 | 0.21/0.35 |
| MoG | 0.11/0.17 | 0.09/0.16 | 0.13/0.20 | 0.15/0.26 | 0.19/0.34 | 0.22/0.37 |
| Coresets | 0.09/0.14 | 0.08/0.14 | 0.10/0.21 | 0.14/0.23 | 0.16/0.28 | 0.20/0.33 |
| GCRL$_{+STGAT}$ | **0.04/0.08** | 0.07/0.12 | 0.12/0.16 | 0.17/0.23 | 0.20/0.29 | 0.23/0.33 |
| GCRL$_{+STGCNN}$ | 0.24/0.43 | 0.30/0.57 | 0.28/0.52 | 0.31/0.49 | 0.35/0.59 | 0.40/0.66 |
| C$^2$INet-online$_{+STGAT}$ | 0.09/0.16 | 0.07/0.13 | 0.10/0.19 | 0.14/0.24 | 0.17/0.29 | 0.20/0.34 |
| C$^2$INet-online$_{+STGCNN}$ | 0.36/0.59 | 0.34/0.54 | 0.25/0.46 | 0.26/0.49 | 0.30/0.55 | 0.34/0.59 |
| C$^2$INet-offline$_{+STGAT}$ | 0.08/0.13 | 0.07/0.12 | 0.11/0.19 | 0.14/0.25 | 0.16/0.27 | 0.18/0.27 |
| C$^2$INet-offline$_{+STGCNN}$ | 0.34/0.55 | 0.31/0.49 | 0.27/0.45 | 0.29/0.51 | 0.28/0.46 | 0.31/0.51 |

Table 3: The table presents the model's average performance across all previously encountered tasks on SDD Dataset. The first column lists the newly added tasks in the continual learning setup, with the preceding colon indicating the average trajectory prediction results for all tasks encountered up to that point. All results are averaged over five runs, with the best outcomes highlighted in bold and the second-best results underlined. Color blocks indicate the ranking of different backbones for comparison.

| TASK | bookstore | :coupa | :deathcircle | :gates | :hyang | :nexus | :little | :quad |
|---|---|---|---|---|---|---|---|---|
| STGAT | 75.96/139.81 | 52.99/98.06 | 114.63/206.93 | 93.75/169.37 | 88.40/163.33 | 79.88/140.89 | 78.35/142.09 | 79.56/145.67 |
| STGCNN | 75.90/139.76 | 52.99/98.14 | 114.97/207.79 | 94.59/172.21 | 91.73/170.78 | 82.79/153.86 | 83.84/157.80 | 81.51/153.27 |
| PECNet | 74.41/137.86 | 51.79/98.23 | 111.03/202.42 | 93.12/170.10 | 88.69/170.23 | 80.54/148.39 | 80.19/151.79 | 82.49/155.83 |
| COUNTERFACTUAL | **66.93/127.97** | **49.84/93.21** | **106.03/194.52** | 90.97/165.97 | 86.50/158.32 | 75.74/135.20 | **75.18/137.10** | 78.06/142.94 |
| INVARIANT | 71.41/132.86 | 52.63/98.05 | 108.44/197.46 | 94.71/170.92 | 90.94/166.05 | 78.69/149.52 | 76.89/149.88 | 80.79/143.42 |
| ADAPTIVE | 86.51/162.14 | 65.21/116.87 | 121.96/216.23 | 99.50/182.85 | 98.32/179.50 | 90.65/173.45 | 94.25/176.25 | 93.32/174.44 |
| EWC | 70.62/131.19 | 54.93/102.42 | 117.01/212.02 | 97.34/176.93 | 96.50/176.75 | 87.44/160.47 | 88.90/163.64 | 88.46/162.84 |
| MoG | 71.31/134.52 | 55.45/104.68 | 118.21/215.39 | 98.14/178.96 | 99.12/181.23 | 90.12/171.26 | 90.56/169.21 | 91.32/169.78 |
| Coresets | 71.25/133.82 | 54.98/102.72 | 117.02/212.03 | 97.35/176.96 | 96.49/176.72 | 87.47/160.51 | 88.93/163.67 | 88.50/162.89 |
| GCRL$_{+STGAT}$ | 76.30/140.42 | 54.87/101.80 | 117.32/212.43 | 97.62/177.25 | 96.58/176.68 | 87.58/160.47 | 89.03/163.62 | 88.63/162.95 |
| GCRL$_{+STGCNN}$ | 76.32/140.44 | 54.87/101.80 | 117.29/212.35 | 97.62/177.25 | 96.57/176.65 | 87.58/160.47 | 89.03/163.62 | 88.63/162.95 |
| C$^2$INet-online$_{+STGAT}$ | 70.87/130.67 | 51.19/95.70 | 107.29/196.40 | 91.56/167.05 | 86.88/161.24 | 75.75/138.72 | 77.21/143.92 | 78.82/143.22 |
| C$^2$INet-online$_{+STGCNN}$ | 70.98/130.84 | 51.50/96.06 | 107.98/196.95 | 91.53/166.96 | 86.86/161.12 | 75.77/140.66 | 77.28/143.96 | 78.82/143.93 |
| C$^2$INet-offline$_{+STGAT}$ | 70.43/130.34 | 50.87/96.04 | 106.68/197.89 | **90.32/164.98** | **85.52/160.31** | **74.42/140.24** | 75.90/142.60 | **77.13/141.69** |
| C$^2$INet-offline$_{+STGCNN}$ | 70.09/130.16 | 50.99/95.98 | 107.01/198.07 | 91.27/166.58 | 85.92/160.95 | 74.98/140.64 | 76.24/142.56 | 78.01/142.85 |

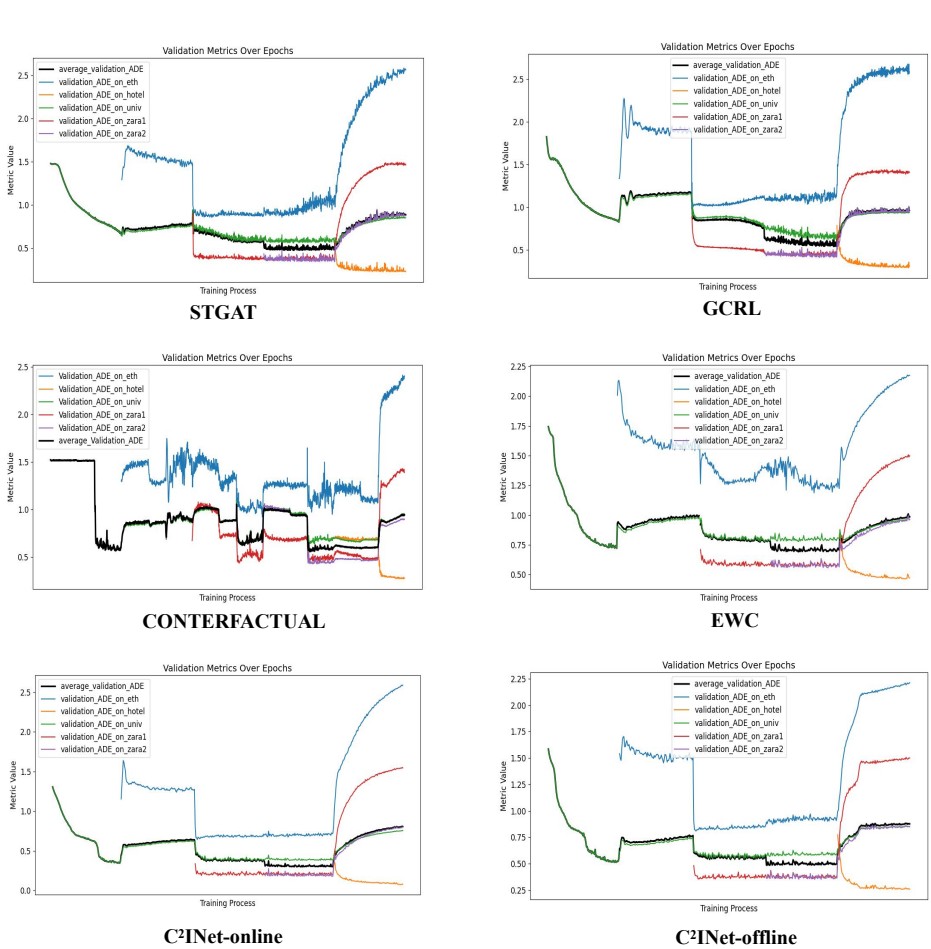

Figure 4: The ADE variations of the validation sets across five task scenarios and their average performance during the training process on the ETH-UCY dataset. The x-axis represents the number of completed training epochs, while the y-axis denotes the corresponding metric values.

optimization disrupting the model's adaptability to previously learned tasks. Although the ADE curves for COUNTERFACTUAL and INVARIANT are relatively stable, their overall performance remains suboptimal.

Fig.6 showcases the performance of various models on the Synthesis dataset. COUNTERFACTUAL and INVARIANT exhibit relatively balanced performance across tasks "0.1" to "0.6", while baseline models such as STGAT and GCRL experience consistent performance degradation, reflecting their tendency to prioritize newly introduced tasks. In contrast, models designed explicitly for continual learning, such as Coresets, EWC, and the proposed $C^2$INet, demonstrate a steady upward performance trend, emphasizing their ability to retain knowledge from previous tasks. In task "0.1", which contains minimal noise, these models maintain strong memory retention, underscoring their robustness in low-noise environments.

In Fig.7, the performance curves on the SDD dataset remain relatively stable across the models. Continual learning models, including $C^2$INet, EWC, and Coresets, effectively mitigate catastrophic forgetting in tasks such as "coupa". However, their performance deteriorates on more challenging tasks like "deathcircle" due to excessive retention of information from earlier tasks. Ultimately, $C^2$INet-offline emerges as the best-performing model, demonstrating superior training performance across the dataset.

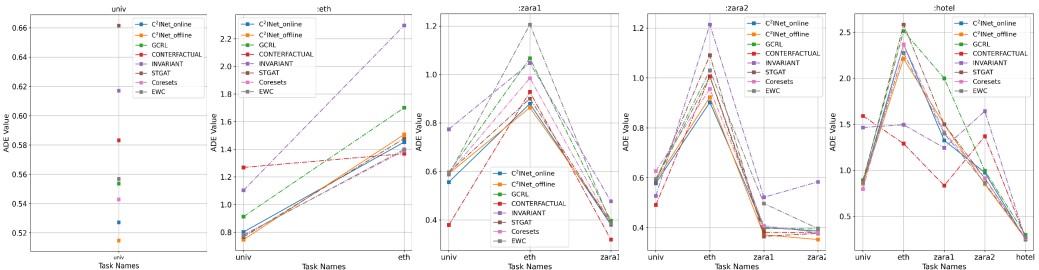

Figure 5: The Average Displacement Error (ADE) on the ETH-UCY dataset for each task, averaged over five runs under continual learning settings. The x-axis represents the sequence of completed tasks, while the y-axis indicates the corresponding metric values.

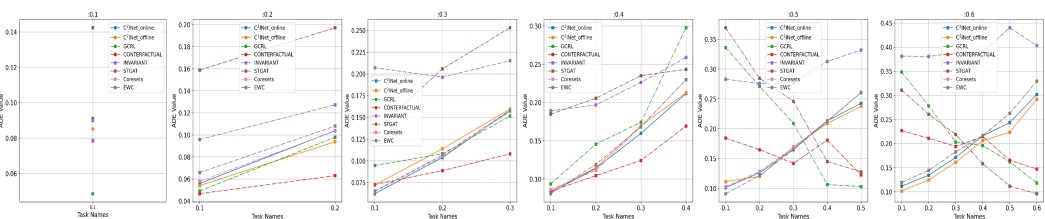

Figure 6: The Average Displacement Error (ADE) on the Synthesis dataset for each task, averaged over five runs under continual learning settings. The x-axis represents the sequence of completed tasks, while the y-axis indicates the corresponding metric values.

### A.4.4 QUALITATIVE RESULTS

Fig.8 mainly displays the qualitative analysis presented in the main text Section 5.5.

### A.4.5 ABLATION STUDY

In Table 4, we systematically assess the contributions of several key modules to the performance of the $C^2$INet model in a continual learning context. The ablation process begins by isolating the impact of the symmetric KL divergence constraint from Eq.10 (Abbreviated as "Divergence"), followed

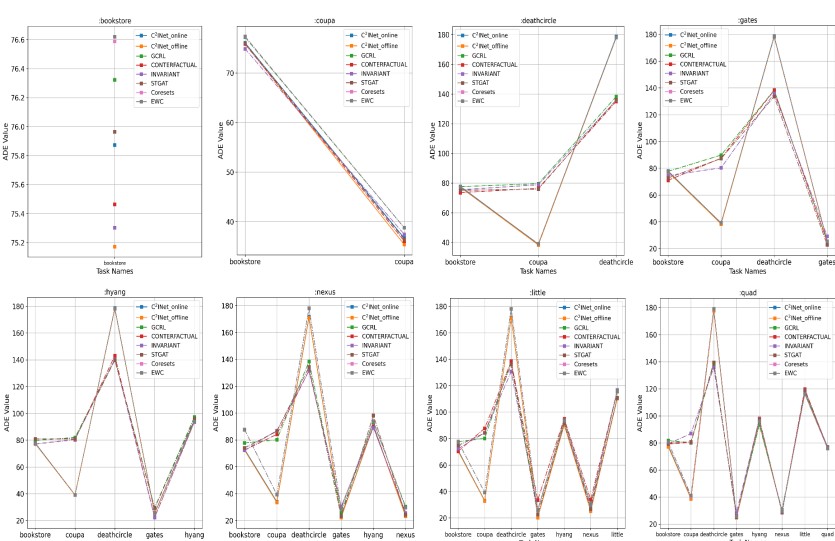

Figure 7: The Average Displacement Error (ADE) on the SDD dataset for each task, averaged over five runs under continual learning settings. The x-axis represents the sequence of completed tasks, while the y-axis indicates the corresponding metric values.

| $C^2I$ | Divergence | Weight | univ | :eth | :zara1 | :zara2 | :hotel |
|---|---|---|---|---|---|---|---|
| ✓ | ✓ | ✓ | **0.820** | **2.228** | **1.420** | **0.827** | **0.247** |
| ✓ | ✓ | ✗ | 0.851(+0.031) | 2.372(+0.144) | 1.574(+0.154) | 0.839(+0.012) | 0.256(+0.009) |
| ✓ | ✗ | ✓ | 0.883(+0.063) | 2.562(+0.334) | 1.663(+0.243) | 0.865(+0.038) | 0.285(+0.038) |
| ✗ | ✗ | ✗ | 0.913(+0.093) | 2.701(+0.473) | 1.752(+0.332) | 0.898(+0.071) | 0.296(+0.049) |

Table 4: Results of the ablation study, focusing on the performance of the $C^2I$Net model on the ETH-UCY dataset after the removal of certain modules. The first row of the table specifies the ablated modules and the completed training tasks. Check mark indicates the removal of the corresponding module, while cross signifies its inclusion.

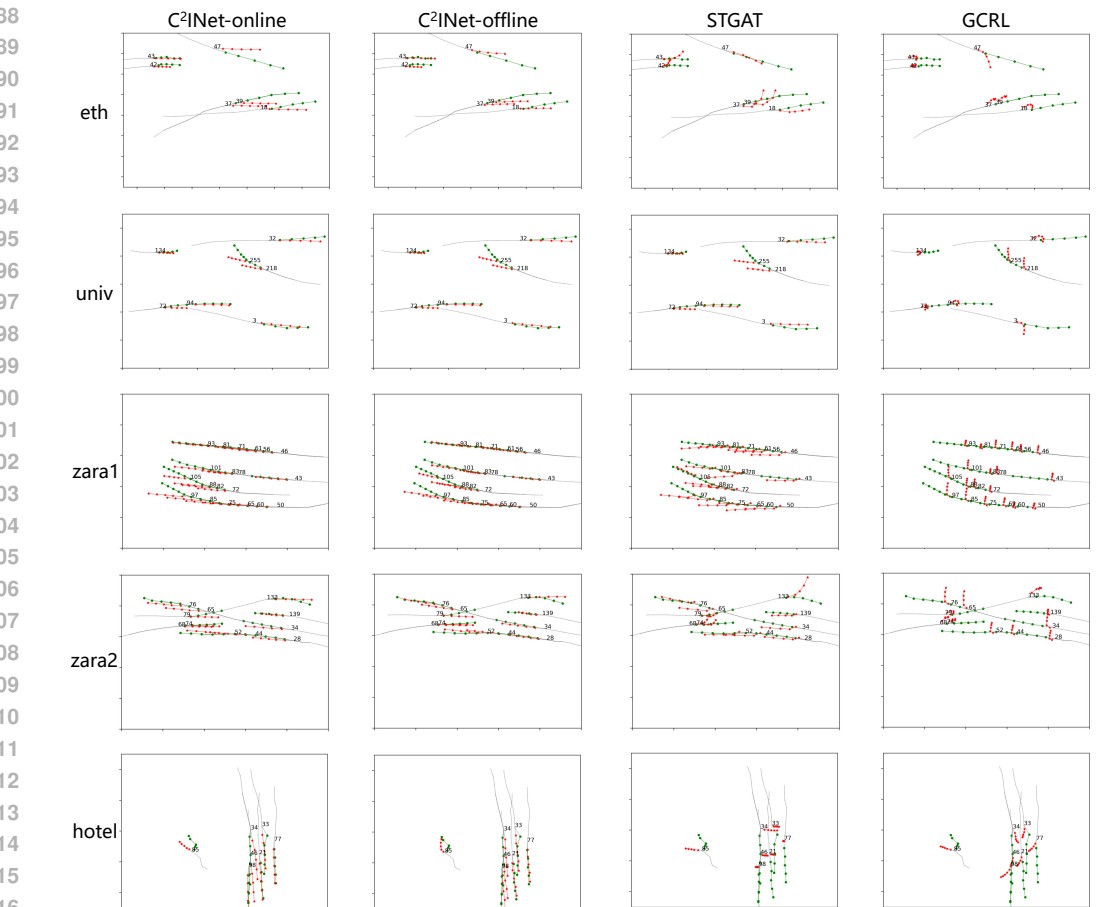

Figure 8: Qualitative results on the ETH-UCY dataset. The gray lines represent observed trajectories, the red points indicate predicted paths, and the green points denote the ground truth.

by the removal of the weight optimization procedure outlined in Eq.9, where only average weight parameters are used (Abbreviated as "Weight"). Lastly, we completely omit the Continual Causal Intervention mechanism to evaluate its significance.

A detailed comparison between the second and third rows of the table demonstrates that when weight optimization is removed and average weights are used, the model's performance initially declines by approximately $3.7\%$ during the early stages of training. However, the model fills the gaps as training progresses and gradually approaches near-optimal performance. This pattern indicates that the lack of weight optimization is mitigated as the prior queue accumulates more elements, diminishing its overall impact on model performance. Conversely, the evident $5\% - 17\%$ performance decline observed after removing the symmetric KL divergence constraint highlights the importance of maintaining diversity among priors. This diversity appears to be crucial in preventing overfitting to specific tasks and ensuring that the model adapts effectively across varying environments.

In the final row, the complete removal of both KL divergence constraints related to confounding factors (from Eq.10), alongside the exclusion of the weight optimization and divergence constraint mechanisms, leads to the most significant performance degradation—up to $21.6\%$. This decline becomes more pronounced as tasks increase, with catastrophic forgetting contributing heavily to the sharp drop in average performance. These results underscore the Continual Causal Intervention module's essential role and the associated constraints in sustaining model robustness during continual learning.

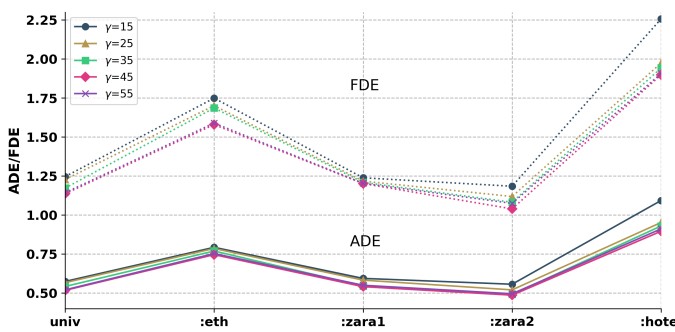

Figure 9: The prediction results of C$^2$INet-online on the ETH-UCY dataset with varying maximum prior queue capacities $\gamma$ are shown. The model regulates the number of priors through a pruning mechanism. Dashed lines represent FDE, while solid lines indicate ADE.

### A.4.6 ANALYSIS OF QUEUE CAPACITY

Fig.9 illustrates the training performance of C$^2$INet-online across varying maximum prior queue capacities $\gamma$. When the number of generated components surpasses $\gamma$ after a task, the proposed pruning mechanism is activated to control the queue size. The experimental results demonstrate that the model achieves optimal performance with $\gamma = 45$, whereas $\gamma = 15$ leads to the poorest outcomes. This finding indicates that selecting an appropriately sized prior queue is crucial for maximizing model efficiency and effectiveness. Both huge and overly constrained prior queues can introduce issues—such as overfitting or reduced learning capacity—highlighting the essential function of the pruning mechanism in maintaining a balance that prevents performance degradation. These results emphasize the importance of reasonably controlled priors to enhance model robustness and adaptability in continual learning environments.

### A.4.7 ANALYSIS OF TASK SEQUENCE

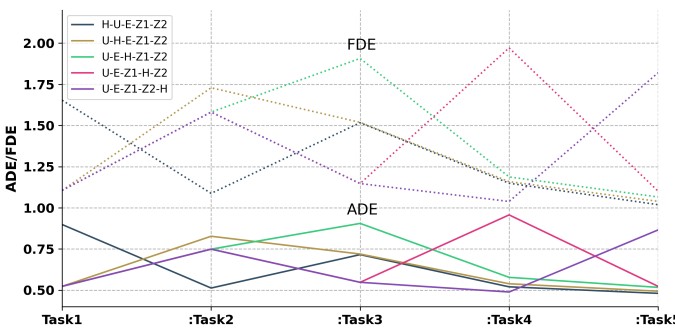

Figure 10: The prediction results of C$^2$INet-online on the ETH-UCY dataset with different task loading sequences are presented. The task mappings are as follows: U - univ, E - eth, Z1 - zara1, Z2 - zara2, H - hotel. Dashed lines represent FDE, while solid lines indicate ADE.

Fig.10 presents the ADE performance of the model across different training task sequences. The results reveal that changes in the order of task training result in subtle but measurable variations in the model's final average performance, with a gap of approximately 7.6% between the best and second-worst outcomes ("H-U-E-Z1-Z2" and "U-E-Z1-H-Z2"). A key observation is that the "hotel" task, which introduces substantial noise, has a noticeable impact on the overall performance across all tasks. Training the "hotel" task earlier in the sequence leads to improved final performance, suggesting that the long-term forgetting effect mitigates the adverse bias caused by the noise. This

finding underscores the impact of task order in continual learning and highlights how forgetting mechanisms can mitigate confounding noise within tasks.

