# OpenReview forum: "C$^{2}$INet: Realizing Incremental Trajectory Prediction with Prior-Aware Continual Causal Intervention"
_ICLR.cc/2025/Conference — Submitted to ICLR 2025_

### Official Review · Reviewer_zjLH · 2024-10-23

**Soundness:** 2
**Presentation:** 1
**Contribution:** 2
**Rating:** 3
**Confidence:** 3

**Summary:**

This paper proposes $C^2INet$ for trajectory prediction, introducing causal intervention to enable continuous trajectory prediction across evolving domains. To address the issue of catastrophic forgetting, they introduce a Continuity Memory Module. Experiments on three datasets demonstrate the effectiveness of the proposed method.

**Strengths:**

1. The paper researches an interesting and critical problem in trajectory prediction.
2. The organization of the paper is somewhat reasonable.

**Weaknesses:**

Some format issue should be considered:
1. The citation format is wrongly used.
2.  the vertical space is not proper set, especially on Page 4
3. Font size of Table 1 is too small, which is weird and does not match the main text

Some Weaknesses about the contents:
1. I’m so confused about the motivation of the work, especially the necessity of introducing causal intervention into the trajectory prediction task.

2. Existing continual learning methods also take into account the catastrophic forgetting problem by utilizing the experience replay mechanism. What are the differences between those CL methods and $C^2INet$? Can your method solve some critical problems that traditional CL methods cannot?

3. The authors use STGAT and STGCNN as backbones. They were proposed in 2016 and 2019, which are now outdated. The method should be integrated with cutting-edge backbones proposed in 2023 and 2024.

4. The results in terms of ADE/FDE on the SDD dataset are too high and seem strange. All results are above 50.0. Do you use a particular coordinate system instead of standard pixel coordinates?

5. Related works are not sufficient. More recent works from the past two years should be incorporated, such as diffusion models [1][2][3] and works with new settings [4][5].

[1]Stochastic Trajectory Prediction via Motion Indeterminacy Diffusion

[2]BCDiff: Bidirectional Consistent Diffusion for Instantaneous Trajectory Prediction

[3]Universal Trajectory Predictor Using Diffusion Model

[4]ITPNet: Towards Instantaneous Trajectory Prediction for Autonomous Driving

[5]Adapting to Length Shift: FlexiLength Network for Trajectory Prediction

**Questions:**

I’m not familiar with causal intervention, so I may not provide professional reviews on the technical part.
Questions for other parts, see weaknesses.

---

> ### Author Response · Authors · 2024-11-17
> **Reply to Reviewer zjLH Part1**
>
> Thanks for your careful and valuable comments. We will explain the concerns point by point.
>
> Q1: Some format issue should be considered.\
> A1: Thank you for pointing this out. Due to page length constraints, some text formats need to be adjusted. We will certainly place greater emphasis on formatting norms. The revised version has corrected issues, including citation format and font size.
>
> Q2: I’m so confused about the motivation of the work, especially the necessity of introducing causal intervention into the trajectory prediction task.\
> A2: I am pleased to further elaborate on our motivation. The generalizability of trajectory prediction tasks is an issue that the community is increasingly focusing on, aiming to achieve a universal prediction model applicable in various practical environments. Causal intervention techniques have been widely used in recent years to eliminate the confounding bias in predictions, enabling trajectory prediction models to learn meaningful features that do not rely on spurious correlations. Some studies have already embarked on this exploration[1][2][3]. However, existing works generally lack consideration for ensuring the continued effectiveness of debiasing mechanisms as causal associations change with the environment. It is also validated in our experiments in section 5.3, indicating that with changes in the environment, the predictive outcomes in past scenarios may decline, and addressing catastrophic forgetting as well as environmental adaptability is a strong starting point for our work.
>
> Q3: Existing continual learning methods also take into account the catastrophic forgetting problem by utilizing the experience replay mechanism. What are the differences between those CL methods and C2INet? Can your method solve some critical problems that traditional CL methods cannot?\
> A3: As you mentioned, continual learning addresses the problem of catastrophic forgetting. Our proposed C2INet preserves the prior information $P(C)$ of continuously changing environments to maintain a long-term ability to eliminate confounding biases originating from the "Prior-Aware" nomenclature. We achieve the memorization of priors through posterior aggregation based on a carefully designed causal intervention structure, which can be referred to in Eq.6-8. The memory of environment-related correlation factors is achieved through optimizable pseudo feature $U$.\
> We also compare our approach with typical continual learning methods (using STGAT as the backbone), including Elastic Weight Consolidation (EWC), which applies gradient constraints; latent features modeled as a mixture of Gaussians with diagonal covariance (MoG); and the random coresets method (Coresets), which uses memory replay during training. Our analysis demonstrates that our approach effectively captures high-quality changes in environmental content, as detailed in section 5.3. Traditional CL methods are indeed effective, but C2INet efficiently identifies the impact of intrinsic factors on trajectory representation. It cleverly integrates continual learning strategies to ensure memory and adaptability under changing training environments. C2INet shows improvements in motivation, technical design, and experimental outcomes over traditional CL methods.
>
> Q4: The authors use STGAT and STGCNN as backbones. They were proposed in 2016 and 2019, which are now outdated. The method should be integrated with cutting-edge backbones proposed in 2023 and 2024.\
> A4: Thank you for pointing this out. Classic never goes out of style. STGAT and STGCNN, as classic RNN-based and CNN-based trajectory representation methods, can simply and effectively demonstrate that our plugin structure can enhance their expressive performance. In fact, other works related to trajectory prediction generalization also adopt the same experimental design to eliminate additional interference, such as GCRL[3] using STGAT as the backbone, [2] using STGAT and PECNET[4], CaDeT[5] using Graph Transformer[6].\
> We also acknowledge your concerns, and therefore, we have included STAR[7] as a backbone for validation on ETH-UCY Dataset, Synthetic Dataset and SDD Dataset (The detailed data results are presented in the table of Part 2) . The reason for not choosing the latest backbone is that the mechanisms in recent years' work are generally complex, making it puzzling to test our plug-and-play capabilities with a basic model. Therefore, we opted for the newer STAR model, which has advantages in handling relationships between agents by Spatio-Temporal Graph Transformer Networks.\
> It can be observed that when using STAR as the backbone, the performance is inconsistent across various datasets, with a notably significant decline on the ETH-UCY dataset. The tested C$^2$INet (trained in online mode) indeed greatly enhances the model's training robustness in different scenarios. More importantly, under the setting of continual learning, the model can efficiently achieve effective representation.

---

> ### Author Response · Authors · 2024-11-17
> **Reply to Reviewer zjLH Part2**
>
> (continue)
>
> ETH-UCY Dataset|STAR|GCRL-STAR|C^{2}Inet-STAR|
> | -------- | -------- | -------- | -------- |
> univ|1.114/1.967|0.832/1.626 | 0.608/1.257|
> :eth   |1.241/ 2.135| 1.065/2.089| 0.819/1.644|
> :zara1 |0.896/1.726 | 0.878/1.745| 0.572/1.227|
> :zara2 |0.671/1.313 | 0.637/1.290| 0.494/1.062|
> :hotel |1.587/2.963 | 0.974/1.971 | 0.510/1.081 |
>
> Synthetic Dataset |STAR | GCRL-STAR  | C^{2}Inet-STAR |
> | -------- | -------- | -------- | -------- |
> 0.1  | 0.069/0.104|0.059/0.104 | 0.0473/0.079 |
> :0.2 |0.085/ 0.144|0.078/0.137| 0.0743/0.127 |
> :0.3 | 0.124/0.195 | 0.114/0.185| 0.1010/0.173|
> :0.4 |0.157/0.249| 0.145/0.238| 0.1319/0.226 |
> :0.5|0.184/0.308|0.177/0.299| 0.1663/0.281 |
> :0.6 |0.225/0.367|0.218/0.359| 0.2062/0.346 |
>
> SDD Dataset| STAR | GCRL-STAR | C^{2}Inet-STAR |
> | -------- | -------- | -------- | -------- |
> bookstore  | 76.236/140.283|75.901/139.901| 75.945/139.768|
> : coupa  | 54.516/101.043| 54.352/101.067| 52.976/98.020|
> : deathcircle| 117.198/212.099|116.429/210.616|115.2337/208.1377  |
> : gates| 97.478/176.860| 96.964/175.811|   94.3827/170.6680    |
> : hyang| 96.455/176.333|96.146/175.680|  91.3278/166.3408  |
> : nexus| 87.426/160.058|87.099/159.347|  82.2640/151.4751   |
> : little| 88.873/163.186| 88.531/162.492|  82.5079/153.4227   |
> : quad| 88.470/162.514| 88.028/161.603|  80.5028/147.8559  |
>
> Q5: The results in terms of ADE/FDE on the SDD dataset are too high and seem strange. All results are above 50.0. Do you use a particular coordinate system instead of standard pixel coordinates?\
> A5: Thank you for paying attention to the details of the experimental result. In fact, we have validated our approach on the SDD dataset using pixel coordinates consistent with other studies, selecting the scene sequence 'Bookstore', 'Coupa', 'Deathcircle', 'Gates', 'Hyang', 'Nexus', 'Little', 'Quad' for continual learning, and focusing on targets such as 'Cart', 'Biker', 'Pedestrian', 'Skater', 'Bus'. We are the first to develop a method that trained on the SDD dataset in a continual learning setting for each scene. Specifically, we sequentially train and test on different scenes in the SDD dataset, with the model accessing only the data samples from the current scene each time. Table 1 in section 5.2 compares the average performance for current and previously completed tasks under continual learning. The experimental results are worse than those obtained with traditional experimental setups, but this is not unreasonable; for example, STGAT can achieve 31.19 ADE under standard settings [14], and in the worst case, it can reach 65.82 ADE in the FEND [15] scenario. Under our experimental setup, it can achieve more than 70 ADE. This is primarily due to insufficient training and catastrophic forgetting when training under different scenes, precisely the issue we aim to address. Of course, the leading advantage of C2INet on the SDD dataset is not clear enough and may not prominently demonstrate our method's advantages. However, it can be observed that many methods fail. To fully illustrate our approach's advantages, we have selected the real-world dataset Apollo dataset[8]. We will supplement the explanation of the results here once all the experimental data have been obtained.

---

> ### Author Response · Authors · 2024-11-26
> **Reply to Reviewer zjLH Part3**
>
> Q6: Related works are not sufficient. More recent works from the past two years should be incorporated, such as diffusion models and works with new settings.\
> A6: Thank you for pointing out these related works. We have supplemented the revised version with descriptions of diffusion models [9][10][11] and works with new settings [12][13] in Appendix A.3.1. These excellent works have also inspired us greatly.
>
> Reference\
> [1] Chen, Guangyi, et al. "Human trajectory prediction via counterfactual analysis." ICCV. 2021.\
> [2] Liu, Yuejiang, et al. "Towards robust and adaptive motion forecasting: A causal representation perspective." CVPR. 2022.\
> [3] Bagi, Shayan Shirahmad Gale, et al. "Generative causal representation learning for out-of-distribution motion forecasting." ICML. PMLR, 2023.\
> [4] Mangalam, Karttikeya, et al. "It is not the journey but the destination: Endpoint conditioned trajectory prediction." ECCV. 2020.\
> [5] Pourkeshavarz, Mozhgan, Junrui Zhang, and Amir Rasouli. "CaDeT: a Causal Disentanglement Approach for Robust Trajectory Prediction in Autonomous Driving."CVPR. 2024.\
> [6] Ziniu Hu, Yuxiao Dong, Kuansan Wang, and Yizhou Sun. Heterogeneous graph transformer. In The World Wide Web Conference, 2020.\
> [7] Yu, Cunjun, et al. "Spatio-temporal graph transformer networks for pedestrian trajectory prediction." ECCV. 2020.\
> [8] Ma, Yuexin, et al. "Trafficpredict: Trajectory prediction for heterogeneous traffic-agents." AAAI. Vol. 33. No. 01. 2019.\
> [9] Gu, Tianpei, et al. "Stochastic trajectory prediction via motion indeterminacy diffusion." CVPR 2022.\
> [10] Li, Rongqing, et al. "Bcdiff: Bidirectional consistent diffusion for instantaneous trajectory prediction." NeurIPS. 2023.\
> [11] Bae, Inhwan, Young-Jae Park, and Hae-Gon Jeon. "SingularTrajectory: Universal Trajectory Predictor Using Diffusion Model." CVPR. 2024.\
> [12] Li, Rongqing, et al. "ITPNet: Towards Instantaneous Trajectory Prediction for Autonomous Driving." SIGKDD. 2024.\
> [13] Xu, Yi, and Yun Fu. "Adapting to length shift: Flexilength network for trajectory prediction." CVPR. 2024.\
> [14] Xu, Chenxin, et al. "Groupnet: Multiscale hypergraph neural networks for trajectory prediction with relational reasoning." CVPR. 2022.\
> [15] Wang, Yuning, et al. "Fend: A future enhanced distribution-aware contrastive learning framework for long-tail trajectory prediction." CVPR. 2023.

---

### Official Review · Reviewer_Nzn5 · 2024-10-30

**Soundness:** 2
**Presentation:** 3
**Contribution:** 2
**Rating:** 5
**Confidence:** 3

**Summary:**

The study introduces C2INet, an innovative method for multi-agent trajectory prediction in intricate settings that utilizes ongoing causal intervention. C2INet integrates causal intervention and continuous learning inside a memory-enhanced framework to tackle challenges such as environmental bias, catastrophic forgetting, and hardware limitations in real-time multi-agent prediction. This method use variational inference to synchronize environment-related priors with a posterior estimator, guaranteeing precise trajectory representation by addressing confounding variables in the latent space.

C2INet's principal innovation is its ongoing learning process, which progressively adapts to new circumstances while maintaining performance on previously encountered ones. This is accomplished via a memory module that retains ideal priors across situations, therefore safeguarding essential knowledge and reducing overfitting via a pruning process. Comprehensive assessments of real-world datasets, including ETH-UCY and Stanford Drone, alongside synthetic datasets, reveal that C2INet surpasses conventional methods in predictive accuracy and resistance to task interference, attaining notable enhancements in metrics such as ADE and FDE across multiple tasks.

C2INet effectively mitigates significant shortcomings in current trajectory prediction models by integrating causal intervention with a continuous memory framework, hence guaranteeing strong performance in dynamic, multi-agent settings.

**Strengths:**

1. Adaptability to Multiple Scenarios:
By leveraging continual causal intervention, C2INet effectively handles confounding factors in diverse scenarios and retains information from previous tasks. This design enhances the model's adaptability to dynamic environments, making it suitable for multi-agent trajectory prediction in complex settings, such as autonomous driving and crowd monitoring.

2. Comprehensive Experimental Validation:
The paper provides extensive validation across multiple datasets (ETH-UCY, Stanford Drone, synthetic datasets) and compares C2INet with various baseline methods, including common causal intervention and continual learning approaches. The results demonstrate that C2INet outperforms traditional methods in key metrics (e.g., ADE and FDE), proving its effectiveness in handling catastrophic forgetting and improving prediction accuracy.

3. Modular Design:
C2INet’s design is modular, making it compatible with multiple baseline models (e.g., STGAT, SocialSTGCNN). This plug-and-play characteristic increases the flexibility of the approach, allowing it to be used in various model architectures and promoting wider applicability.

**Weaknesses:**

1. Innovative Design:
The paper introduces a novel model, C2INet, which combines causal intervention with continual learning, specifically using a memory module to retain optimal priors across different scenarios to mitigate catastrophic forgetting. This approach is relatively rare in multi-task learning and offers a degree of originality.

2. Adaptability to Multiple Scenarios:
By leveraging continual causal intervention, C2INet effectively handles confounding factors in diverse scenarios and retains information from previous tasks. This design enhances the model's adaptability to dynamic environments, making it suitable for multi-agent trajectory prediction in complex settings, such as autonomous driving and crowd monitoring.

3. Potential Limitations of the Prior Memory Module:
Although the prior memory module helps alleviate catastrophic forgetting, it heavily relies on storing priors for different tasks, which may lead to challenges in memory management and capacity. As the number of tasks grows, the memory module might struggle to scale efficiently. Additionally, the paper does not discuss how to effectively manage priority or memory compression when storage space is limited.

4. Limitations of the Experimental Datasets:
Although the paper uses multiple datasets, these are mainly focused on specific domains (e.g., pedestrian and vehicle trajectory prediction) and lack diversity. The generalizability of the experimental results to more complex or diverse dynamic environments is unclear, limiting the method’s applicability to real-world scenarios.

5. Complex Hyperparameter Tuning with Limited Guidance:
C2INet involves multiple key hyperparameters (e.g., the KL divergence adjustment coefficient, weights in the memory module) that significantly affect model performance, but the paper does not provide detailed guidance on tuning them. The complexity of hyperparameter tuning, combined with a lack of explicit guidelines, may hinder other researchers from reproducing and applying the method in different settings.

**Questions:**

You have not provided detailed mathematical derivations for the causal intervention section. Could you elaborate further or reference additional causal inference theories to support the method? Specifically, how do you ensure that the intervention effectively removes confounding factors, and have you considered potential causal relationships between different tasks?

What are the computational complexity and storage requirements of C2INet? How do you ensure real-time performance in hardware-constrained environments (e.g., embedded systems or mobile devices)? Is there a quantitative analysis of resource consumption, or have you tried to optimize the algorithm to reduce computational load?

As the number of tasks increases, will the storage requirements for the prior memory module become unmanageable? How do you manage priorities or compress the memory module effectively when storage space is limited? Have you considered the issue of memory overflow as the number of tasks continues to grow?

C2INet involves several key hyperparameters (e.g., the KL divergence adjustment coefficient, weights in the memory module). Could you provide insights on how to tune these hyperparameters? Have you considered an adaptive hyperparameter optimization mechanism to reduce the reliance on manual tuning?

The datasets used are mainly focused on pedestrian and vehicle trajectory prediction, which is somewhat limited in scope. Do you plan to test C2INet in more diverse and complex scenarios to evaluate its applicability and generalizability?

Given that C2INet includes multiple complex modules (e.g., memory module and causal intervention), how interpretable is the model? Do you plan to provide more intuitive visualizations or explanations to demonstrate the model’s decision-making process?

Since the paper has already discussed the situation of hardware resource constraints, has it considered other optimization methods such as hardware acceleration or knowledge distillation? What advantages does the proposed optimization method have compared to these alternative approaches?

---

> ### Author Response · Authors · 2024-11-18
> **Reply to Reviewer Nzn5 Part1**
>
> Thank you for reviewing and evaluating our work. We are especially grateful for your recognition of our Innovative Design, Adaptability, and Experiments. Overall, we have made additions in accordance with your requests, including supplementary mathematical derivations, computational resource analysis, and the addition of new experimental datasets.
> Next, we will specifically address the several concerns you have raised:
>
> Q1: You have not provided detailed mathematical derivations for the causal intervention section. Could you elaborate further or reference additional causal inference theories to support the method? Specifically, how do you ensure that the intervention effectively removes confounding factors, and have you considered potential causal relationships between different tasks?\
> A1: We apologize for the confusion caused by the previously incomplete derivation details. We have supplemented the appendix A.2 with the derivations of the causal intervention in section 2.3 and the objective function in section 3. Overall, we derive $P(Y|do(X))$, the conditional probability relationship between the confounding factor $C$ and the observed variable $X$, based on two rules of do-calculus: \emph{Action/observation exchange} and \emph{Insertion/deletion of actions}. Essentially, the above debiasing process aims to eliminate spurious correlations. This is accomplished by cutting off the path $C \rightarrow X$ and establishing the direct relationship $X \rightarrow Z \rightarrow Y$. However, environmental influences are often inaccessible in real-world scenarios, and collecting comprehensive trajectory data is costly and complex. The crux of accomplishing debiasing is to obtain an accurate distribution of the confounding factors $P(C)$, which we achieve here through the posterior estimators of samples for precise approximation. The visualized experimental results in Section 5.6 confirm our ability to accurately obtain separable environmental priors for each task. As you mentioned, even with different tasks, causal relationships may exist. We obtain the priors of continuously introduced different tasks and their reasonable weights through optimization. This process ensures that the aggregated priors via the encoder can serve as an effective feature space of confounders for the debiasing process during training.
>
> Q2: What are the computational complexity and storage requirements of C2INet? How do you ensure real-time performance in hardware-constrained environments (e.g., embedded systems or mobile devices)? Is there a quantitative analysis of resource consumption, or have you tried to optimize the algorithm to reduce computational load?\
> A2: Thank you for your constructive comments. We have added an analysis of the model's computational and storage requirements. Our C2INet, to adapt to continuous training scenarios, offers both online and offline training strategies. The online mode is more suitable for real-world applications, enabling rapid training updates. During the training process for each scenario, after a set number of batch rounds, a new component's pseudo feature set is trained and optimized according to Eq.8 of the revised paper, followed by iterative optimization of its optimal weights using Eq.9. Notably, the above process is interspersed within the regular training of the model (it can even be carried out in parallel, a fact we will verify in the future), with the additional computational resource cost being only the training of the component. The storage requirement is mainly used to store the pseudo feature set in the memory module. The offline mode involves unified local offline training after collecting a large amount of data, with the volume of parameter optimization being a multiple related to the memory capacity compared to the online mode. However, the computational load is also utterly acceptable in real-world applications due to the limited queue capacity. \
> Below, we have documented the costs incurred by backbone and C2INet, including training duration, Flops, number of parameters, and storage amount. The inference process incurs no additional overhead. It can be seen that C2INet does not bring significant additional overhead to the regular training of the Backbone model, whether in terms of computation or storage.
> | Index | STGAT（Backbone）| C2INet |
> | -------- | -------- | -------- |
> | Time | 5938ms/Epoch | 2.98ms/Epoch |
> | Flops | 74.73 M/Epoch | 7056/Epoch |
> | Parameters | 4234 | 3664 |
> | Storage | 402192B | 34B |

---

> ### Author Response · Authors · 2024-11-18
> **Reply to Reviewer Nzn5 Part2**
>
> Q3: As the number of tasks increases, will the storage requirements for the prior memory module become unmanageable? How do you manage priorities or compress the memory module effectively when storage space is limited? Have you considered the issue of memory overflow as the number of tasks continues to grow?\
> A3: Thank you for raising this important issue. Storage requirements are an essential consideration for edge devices; hence, in the article, we thoroughly considered how to maintain component expansion within a controllable range while preserving maximal diversity. For specifics, you can refer to Page 6, Line 316 of the initial version of the paper. Based on the set memory capacity, we prune components with similar information; the pruning strategy involves calculating the two most similar prior components by the similarity $S(\cdot,\cdot)$ and removing the one that contributes the least to the overall information content. The related process can also be found in Line 21 of Algorithm 1 in Appendix A.1: pruning is performed when the queue length exceeds $\gamma$. We hope the above can address your concerns.
>
> Q4: C2INet involves several key hyperparameters (e.g., the KL divergence adjustment coefficient and weights in the memory module). Could you provide insights on how to tune these hyperparameters? Have you considered an adaptive hyperparameter optimization mechanism to reduce the reliance on manual tuning?\
> A4: Thank you for your question. It may be that our method description is not detailed enough. We have added the derivation process in the revised appendix to address this issue. Moreover, the training optimization process can be found in the algorithm description in Appendix A.1. The only parameter that requires manual adjustment throughout the training process is the capacity of the memory module, $\gamma$, for which we have designed experiments and provided explanations in Appendix A4.4.\
> The several vital hyperparameters you mentioned can be involved in the optimization process. Briefly, the overall training process is conducted through a min-max mechanism. In the minimization step, building on the convex property of Eq.5, we alternately optimize two variables: $P_{K}(C)$, representing the shift in the prior probability density function, and $\alpha_k$, which adjusts the scaling factor. The first optimization aims to bring the prior closer to the aggregated posterior probability while maintaining as much inconsistency as possible with the content $M_{\leq K-1}(C)$ (Eq.6) already obtained in the queue. Once the prior $P_{K}(C)$ is determined and fixed, the weights of the components can be optimized according to application requirements either online (Line 290) or offline (Line 303) modes.
>
> Q5: The datasets used are mainly focused on pedestrian and vehicle trajectory prediction, which is somewhat limited in scope. Do you plan to test C2INet in more diverse and complex scenarios to evaluate its applicability and generalizability?\
> A5: Thank you for pointing out the issues above. We have supplemented with the real-world driving scenario dataset Apollo[1] for validation. The Apollo trajectory prediction dataset comprises 53 one-minute vehicle trajectory segments, each with a sampling rate of 2 frames per second. The categories of agents include small vehicles, big vehicles, pedestrians, motorcyclists, bicyclists, and others. Based on the number of agents present in each segment, the dataset can be divided into five scenarios:\
> Scene 1 (Very few targets): [0, 40), 3 segments\
> Scene 2 (Few targets): [40, 70), 15 segments\
> Scene 3 (Moderate number of targets): [70, 100), 14 segments\
> Scene 4 (Many targets): [100, 140), 13 segments\
> Scene 5 (Very many targets): [140, +\infty),  8 segments\
> The experimental results are shown in the table below.
> |Index | STGAT | STGCNN | GCRL-STGAT | GCRL-STGCNN |C$^{2}$Inet-STGAT|C$^{2}$Inet-STGCNN|
> | -------- | -------- | -------- | -------- | -------- | -------- | -------- |
> |Scene 1| 5.961/10.094| 6.261/10.699 | **5.737/9.694** | **6.04/10.32** |6.136/10.319 | 6.088/10.240 |
> |:Scene 2 | 6.018/10.193 | 6.430/10.854 | 5.247/9.09 | 5.001/8.43 |**2.840/4.822** |**2.928/4.47**|
> |:Scene 3 | 3.478/5.517 | 7.332/12.749 | 2.408/4.116 | 3.949/6.951 | **2.223/3.790** |**2.181/3.780**|
> |:Scene 4 | 2.823/4.572 | 6.470/11.053 | 2.607/4.292 | 3.327/5.672 | **2.445/4.082** |**2.201/3.812**|
> |:Scene 5 | 2.906/4.881 | 4.277/7.486 | 2.673/4.613 | 3.007/5.075 | **2.590/4.415** |**2.116/3.592**|
>
> It can be discerned that STGAT's representational capacity is superior to that of STGCNN. Particularly, STGCNN demonstrates inferior performance as the task duration increases. GCRL, with the introduction of causal intervention, can enhance the model's capabilities in various contexts, but it does not achieve optimal performance during prolonged training. C2INet effectively improves training stability, with the model's performance remaining stable from the second scenario onwards.

---

> ### Author Response · Authors · 2024-11-18
> **Reply to Reviewer Nzn5 Part3**
>
> Q6: Given that C2INet includes multiple complex modules (e.g., memory module and causal intervention), how interpretable is the model? Do you plan to provide more intuitive visualizations or explanations to demonstrate the model’s decision-making process?\
> A6: Thank you for highlighting the issues above. We have optimized the model framework diagram in Fig.1 of the initial version of the paper, which will be presented in the revised Fig.2 (to be uploaded shortly). In the upper part of the figure, we describe the optimization process of Causal Intervention, which introduces environment-related priors through variational inference and cuts off the influence of spurious features on the correlation with trajectory data. However, the debiasing capability of the past becomes ineffective with changes in the environment. To counteract catastrophic forgetting, we seamlessly integrate the proposed memory module with the causal intervention mechanism and ensure low resource occupancy through a pruning strategy. This allows the environmental context encoding to always obtain a continuously optimized confounding prior through multiple pseudo features. Our method is primarily used to enhance training effectiveness. For inference, it is achieved directly through the normal trajectory decoding process.
>
> Q7：Since the paper has already discussed the situation of hardware resource constraints, has it considered other optimization methods such as hardware acceleration or knowledge distillation? What advantages does the proposed optimization method have compared to these alternative approaches?\
> A7：The issue you raised is precisely what we have considered. We have considered several aspects of enhancing overall efficiency during the training process. \
> Firstly, we provide dynamic memory queue capacity, allowing dynamic control of storage resource consumption based on changes in actual scenarios. We have tested queue capacity in Appendix A.4.4.\
> Secondly, we have considered both online and offline versions for training pseudo features. The online mode updates only one component and its weight value at a time, which significantly improves computation speed and is more suitable for edge devices. We have analyzed the FLOPs and the number of parameters saved for each training instance in Q2. Indeed, our prior intervention enhances the training effect of the trajectory prediction model as a plug-in and supports various hardware acceleration methods. As an additional constraint, it can even be conducted in parallel with the primary model's training process, significantly improving its efficiency (related experimental results will be released in the future). At the same time, we implement hardware resource optimization using an information gain-based pruning strategy.\
> Third is the generation frequency (the number of batch intervals between two pseudo features training) which affects training speed. In the original article, we set this to be fixed. To verify whether the generation frequency affects the results, we test it on the ETH-UCY dataset. We control the number of epochs for each task to be 300, with a memory queue capacity of 40.  It is observed that the expansion rate of components does not significantly affect model performance. However, the model performs better at an appropriate expansion rate. Therefore, a comprehensive consideration of the balance between training efficiency and model performance should be taken into account in practical applications.\
>
> |Generation Frequency | Univ |  eth |          zara1 |           zara2 |     hotel |
> | -------- | -------- | -------- | -------- | -------- | -------- |
> |4 | 0.558/1.217 | 0.771/1.724 | 0.581/1.244  | 0.535/1.175 | 1.029/2.248|
> |6| 0.569/1.228 | 0.781/1.714 | 0.581/1.214  |  0.525/1.115 | 0.978/1.947|
> |8| 0.528/1.187 | 0.762/1.686 | 0.551/1.193  |  0.485/1.065 | 0.919/1.898|
> |10| 0.538/1.197 | 0.769/1.704 | 0.560/1.203 |   0.495/1.095| 0.929/1.908|
> |12|0.548/1.207|0.774/1.708|0.561/1.223|0.499/1.105|0.939/1.934|
>
> As you mentioned, traditional methods such as knowledge distillation cannot effectively solve the problem of catastrophic forgetting under continual learning, which is precisely what we are dedicated to resolving.
>
> Reference\
> [1]Ma, Yuexin, et al. "Trafficpredict: Trajectory prediction for heterogeneous traffic-agents." AAAI. 2019.

---

### Official Review · Reviewer_Bhsa · 2024-11-04

**Soundness:** 2
**Presentation:** 3
**Contribution:** 3
**Rating:** 6
**Confidence:** 3

**Summary:**

The paper introduces C2INet, a novel model designed to enhance multi-agent trajectory prediction in dynamic environments by addressing key challenges like environmental bias and catastrophic forgetting. C2INet incorporates a prior-aware memory module that stores optimal priors across tasks, enabling it to adapt incrementally to new scenarios while retaining performance on past tasks. A core component of the model is the continual causal intervention mechanism, which aims to disentangle latent confounding factors that could negatively impact prediction accuracy.

The model's design emphasizes a balance between computational efficiency and performance, employing variational inference to align environment-dependent priors with posterior estimates, thus maintaining robustness against latent biases. The inclusion of a pseudo-feature-based pruning strategy ensures the memory module remains efficient and manageable even as task volume increases. C2INet's framework is evaluated on datasets such as ETH-UCY and Stanford Drone, showcasing its strong adaptability and significant improvements in prediction metrics like Average Displacement Error (ADE) and Final Displacement Error (FDE) when compared to traditional trajectory prediction models.

**Strengths:**

The paper presents a novel method combining causal intervention with a continual learning framework, effectively addressing the problem of bias and forgetting in multi-task learning.

The experimental analysis demonstrates that C2INet performs robustly on multiple datasets, achieving significant improvements in trajectory prediction accuracy compared to existing methods.

The integration of the memory module and prior queue is well-motivated and effectively implemented to handle changing environments, showcasing the potential for real-world applications.

**Weaknesses:**

Although the paper introduces causal intervention comprehensively, some theoretical explanations, particularly regarding the optimization of KL divergence and multi-task prior adjustments, could benefit from simplification. This would make the paper more accessible to a broader audience.

The experiments are thorough but somewhat limited in terms of application diversity. The paper could be strengthened by including analyses on more varied or complex real-world scenarios, such as real-time predictions in live driving conditions.

Certain sections, such as the derivation of equations and framework details, are presented in a complex manner that may challenge the reader's understanding. A clearer and more concise explanation would enhance readability.

While the paper mentions addressing environmental biases, there is insufficient analysis of other types of potential biases in the dataset, such as sample distribution imbalance and long-tail effects. The paper does not delve deeply into how the model performs in broader contexts.

**Questions:**

Could the authors elaborate on how the proposed C2INet model could be implemented in practical real-world systems, especially where computational resources may be limited (e.g., embedded or low-power devices)?

Can the authors provide a more simplified explanation of the theoretical basis, particularly the optimization of KL divergence and the adjustment of priors in multi-task scenarios? This would help make the paper more accessible to a broader audience.

Is there any plan to test the generalization ability of C2INet on larger or more diverse datasets, including scenarios with real-time prediction in live driving conditions? How might the model perform in unseen or significantly different environments?

While the paper discusses mitigating environmental bias, how does C2INet handle other types of data biases, such as sample distribution imbalances or long-tail effects? Would these impact the model's performance or lead to biased predictions?

Could the authors provide more details about the computational complexity of C2INet, such as the number of parameters and resource usage? This would clarify the scalability and practicality of the model for large-scale or real-time applications.

---

> ### Author Response · Authors · 2024-11-18
> **Reply to Reviewer Bhsa Part1**
>
> Thank you very much for your positive feedback. Based on your comments, we will further refine our work, and now I will address the issues you have raised.
>
> Q1: Could the authors elaborate on how the proposed C2INet model could be implemented in practical real-world systems, especially where computational resources may be limited (e.g., embedded or low-power devices)?\
> A1: Thanks for your constructive question. The goal of C2Inet is to be applied in real-world scenarios. Our mechanism is akin to a plug-in designed to enhance the training effectiveness of trajectory prediction models, focusing primarily on optimizing the continuously added priors $\hat{P}^{*}$ and their weights $\alpha$ in variational inference. We have proposed two training modes for different application scenarios: online and offline. The online mode is selected when device resources are limited or access to the entire dataset is limited. In hardware devices, only a tiny amount of storage space is needed to store the task-related prior components that are continuously iterated (our proposed pruning strategy can ensure this).\
> Moreover, optimizing a few parameters will not require too many computational resources. This mode supports the actual vehicle and continuously optimizes the model's performance in collecting extensive road data, which significantly benefits real-world application scenarios. In our future work, we will also attempt to validate it on actual roads. The offline strategy involves unified local offline training after collecting complete data. Its parameter quantity is multiple, which relates to memory capacity compared to the online mode. However, due to the limited queue capacity, its computational load is also completely acceptable in real-world applications.
>
> Q2: Can the authors provide a more simplified explanation of the theoretical basis, particularly the optimization of KL divergence and the adjustment of priors in multi-task scenarios? This would help make the paper more accessible to a broader audience.\
> A2: Thank you for your feedback. We apologize for the previous explanation not being comprehensive enough. We have supplemented the derivations of the causal intervention and the objective function in Appendix A.2 of the revised version.\
> Briefly, our goal is to maximize the probability density $P(Y|do(X))$, where do-calculus is used to intervene in the confounding factors affecting the observed sample $X$. To facilitate the calculation of confounding $P(C)$ in the environment, we introduce variational inference to approximate the actual posterior distribution of latent variables. We propose an additional context encoder $Q(C|X)$ to estimate $C$. To achieve continuous learning capability, we aim to obtain the optimal prior $P(C)$ that can satisfy the training needs of different environments inspired by research related to priors in the field of variational inference. In practice, we subtly take the characteristic distribution of confounding factors as the optimization goal, providing the objective function for multiple environments and its formula transformations (Eq.2, Eq.4-6, and Eq.18). Furthermore, by employing limit approximation and increasing information entropy, we can derive the formula Eq.8 for calculating the new component. Based on the bi-convexity property, we sequentially update the pseudo feature $U_K$ of the newly added component and its corresponding coefficient $\alpha_{K-1}$.
>
> Q3: Is there any plan to test the generalization ability of C2INet on larger or more diverse datasets, including scenarios with real-time prediction in live driving conditions? How might the model perform in unseen or significantly different environments?\
> A3: Thank you for noting this. Our research is aimed at addressing practical application needs and will continue to incorporate data from real-world scenarios to validate our methods. Here, we supplement the Apollo[1] dataset for validation, comparing their performance across different methods. The Apollo trajectory prediction dataset comprises 53 one-minute vehicle trajectory segments, each with a sampling rate of 2 frames per second. The categories of agents include small vehicles, big vehicles, pedestrians, motorcyclists, bicyclists, and others. Based on the number of agents present in each segment, the dataset can be divided into five scenarios:\
> Scene 1 (Very few targets): [0, 40), a total of 3 segments\
> Scene 2 (Few targets): [40, 70), a total of 15 segments\
> Scene 3 (Moderate number of targets): [70, 100), a total of 14 segments\
> Scene 4 (Many targets): [100, 140), a total of 13 segments\
> Scene 5 (Very many targets): [140, +\infty), a total of 8 segments\
> The experimental results are shown in the table below.

---

> ### Author Response · Authors · 2024-11-18
> **Reply to Reviewer Bhsa Part2**
>
> (continue)
> |Index | STGAT | STGCNN | GCRL-STGAT | GCRL-STGCNN |C$^{2}$Inet-STGAT|C$^{2}$Inet-STGCNN|
> | -------- | -------- | -------- | -------- | -------- | -------- | -------- |
> |Scene 1| 5.961/10.094| 6.261/10.699 | **5.737/9.694** | **6.04/10.32** |6.136/10.319 | 6.088/10.240 |
> |:Scene 2 | 6.018/10.193 | 6.430/10.854 | 5.247/9.09 | 5.001/8.43 |**2.840/4.822** |**2.928/4.47**|
> |:Scene 3 | 3.478/5.517 | 7.332/12.749 | 2.408/4.116 | 3.949/6.951 | **2.223/3.790** |**2.181/3.780**|
> |:Scene 4 | 2.823/4.572 | 6.470/11.053 | 2.607/4.292 | 3.327/5.672 | **2.445/4.082** |**2.201/3.812**|
> |:Scene 5 | 2.906/4.881 | 4.277/7.486 | 2.673/4.613 | 3.007/5.075 | **2.590/4.415** |**2.116/3.592**|
>
> It can be discerned that STGAT's representational capacity is superior to that of STGCNN. Particularly, STGCNN demonstrates inferior performance as the task duration increases. GCRL, with the introduction of causal intervention strategies, can enhance the model's capabilities in various contexts, but it does not achieve optimal performance during prolonged training. C2INet effectively improves training stability, with the model's performance remaining stable from the second scenario onwards.
>
> Q4: While the paper discusses mitigating environmental bias, how does C2INet handle other types of data biases, such as sample distribution imbalances or long-tail effects? Would these impact the model's performance or lead to biased predictions?\
> A4: As you mentioned, the long-tail effect and class imbalance issues significantly impact the performance of trajectory prediction models. Indeed, our method can mitigate these issues to some extent.\
> First, our training process iterates sequentially across different scenarios, where the class imbalance is naturally part of the confounding factors. In each scenario, various long-tail distribution issues are prevalent, such as feature-level imbalance highlighted by Fend[2], and trajectory length imbalance pointed out by FlexiLength[3]. Ablation experiments in Appendix A.4.3 demonstrate that the causal intervention mechanism can improve model performance, which also addresses the long-tail distribution issue. In the future, we will further investigate the impact of long-tail distribution issues on environmental confounding. Secondly, our offline mode obtains the category distribution of sample features through pre-clustering at the beginning of task training, while the online mode gradually forms category clusters. Both of these methods effectively resolve class imbalance, as visualized in the confounding prior in section 5.6.
>
> Q5：Could the authors provide more details about the computational complexity of C2INet, such as the number of parameters and resource usage? This would clarify the scalability and practicality of the model for large-scale or real-time applications.\
> A5: Thank you for pointing out considerations regarding practical applications. Our C2INet offers both online and offline training modes, with the online strategy being more suitable for real-world application scenarios, enabling rapid training. In each new scenario, after a set number of training batch rounds, a new component's pseudo feature set is added and optimized according to Eq.8 in the revised version, followed by iterative optimization of its optimal weights using Eq.9. Notably, the above process is interspersed within the model's regular training routine, with the additional computational resource cost being only the training of the component. The storage cost is the retention of the obtained pseudo feature set in the memory module.\
> Below, we have recorded the normal training and the additional overhead brought by CI2Net, including training time, Flops, number of parameters, and storage amount. There is no additional overhead during the inference process. C2INet does not impose a significant additional resource burden on the training of the existing model.
>
> Reference\
> [1]	Ma, Yuexin, et al. "Trafficpredict: Trajectory prediction for heterogeneous traffic-agents." AAAI. 2019.\
> [2]	Wang, Yuning, et al. "Fend: A future enhanced distribution-aware contrastive learning framework for long-tail trajectory prediction." CVPR. 2023.\
> [3]	Xu, Yi, and Yun Fu. "Adapting to length shift: Flexilength network for trajectory prediction." CVPR. 2024.

---

### Author Response · Authors · 2024-11-25
**Reply to all**

We appreciate the valuable comments from the area chairs and the reviewers. Overall, we have addressed and made adjustments according to the issues you have raised as follows:\
-We have supplemented the experimental validation, including additional datasets and backbone.\
-We have expanded on the details of the methods, including the analysis and mathematical derivation of causal inference and optimization objectives.\
-We have added an analysis of computational and storage resources, demonstrating the feasibility of practical applications.\
The related modifications have been incorporated into the revised version and are highlighted in blue. Additionally, we have responded to all questions and welcome ongoing discussions with you.

---

### Meta-Review · Area_Chair_gNez · 2024-12-23

**Metareview:**

This paper proposes  a model for multiagent trajectory prediction. The key innovation centers around causal intervention for continuous trajectory prediction in dynamic environments.

The biggest criticism that is common across all the reviews is lack of clarity in exposition. Most of the reviewers ask for simplified presentation and crisp descriptions. Additionally there are questions about the experimental evaluation.

My recommendation is based upon the fact that we really need to see a whole new manuscript to see if it really addresses the presentation concerns.

**Additional Comments On Reviewer Discussion:**

As mentioned above there are common themes across the reviewers.

---

### Decision · Program_Chairs · 2025-01-22

Reject